

# On the representation of major stratospheric warmings in reanalyses

Blanca Ayarzagüena[1,2], Froila M. Palmeiro[1a], David Barriopedro[2], Natalia Calvo[1], Ulrike Langematz[3] and Kiyotaka Shibata[4]

[1]Dpto. Física de la Tierra y Astrofísica, Universidad Complutense de Madrid, Madrid, 28040, Spain
[2]Instituto de Geociencias (IGEO), Consejo Superior de Investigaciones Científicas - Universidad Complutense de Madrid, Madrid, 28040, Spain.
[3]Institut für Meteorologie, Freie Universität Berlin, Berlin, 12165, Germany.
[4]School of Environmental Science and Engineering, Kochi University of Technology, Kami, 7828502, Japan.

[a] now at: Barcelona Supercomputing Center (BSC-CNS)

*Correspondence to*: Blanca Ayarzagüena (bayarzag@ucm.es)

**Abstract.** Major sudden stratospheric warmings (SSWs) represent one of the most abrupt phenomena of the boreal wintertime stratospheric variability, and constitute the clearest example of coupling between the stratosphere and the troposphere. A good representation of SSWs in climate models is required to reduce their biases and uncertainties in future projections of stratospheric variability. The ability of models to reproduce these phenomena is usually assessed with just one reanalysis. However, the number of reanalyses has increased in the last decade and their own biases may affect the model evaluation. Here we compare the representation of the main aspects of SSWs across reanalyses. The examination of their main characteristics in the pre- and post-satellite periods reveals that reanalyses behave very similarly in both periods. However, discrepancies are larger in the pre-satellite period than afterwards, particularly for the NCEP/NCAR reanalysis. All datasets reproduce similarly the specific features of wavenumber-1 and wavenumber-2 SSWs. A good agreement among reanalyses is also found for triggering mechanisms, tropospheric precursors and surface fingerprint. In particular, differences in blocking precursor activity of SSWs across reanalyses are much smaller than between blocking definitions.

## 1 Introduction

Major sudden stratospheric warmings (SSWs) constitute the most important phenomena of the Northern Hemisphere polar stratospheric variability in wintertime. They are abrupt warmings of the polar stratosphere that lead to a deceleration of the polar vortex and a reversal of the typical westerly circulation (Andrews et al., 1987). SSWs can be classified into two different types according to the structure of the polar vortex during the event. Accordingly, the polar vortex is either displaced from the polar cap (vortex displacement, D SSWs) or split into two parts of similar size (vortex split, S SSWs) (Labitzke and Naujokat, 2000).

SSWs represent a clear example of stratosphere-troposphere coupling in both directions. First, they are usually preceded by an enhancement of upward-propagating wave activity (e.g. Matsuno, 1971). The sources of this anomalous wave activity are



mainly located in the troposphere and correspond to anomalous circulation events such as blocking highs, among others (e.g. Martius et al., 2009; Nishii et al., 2011; Ayarzagüena et al., 2011; Barriopedro and Calvo, 2014). Based on the wave activity preceding SSWs, they are commonly classified into wavenumber 1 (WN1) or wavenumber 2 (WN2) events (e.g. Bancalà et al., 2012; Barriopedro and Calvo, 2014). This classification produces subsets of events similar to the D/S catalogue. However,

there are differences since the former is based on the precursory wave activity while the D/S classification accounts for the shape of the polar vortex during the post-warming phase (Bancalà et al., 2012). Depending on the type of SSWs, the tropospheric precursors are different and/or located in different geographical locations (Cohen and Jones, 2011; Bancalà et al., 2012). In particular, differences in blocking precursors are larger when SSWs are classified into WN1/WN2 than D/S (Barriopedro and Calvo 2014).

In terms of downward coupling, SSWs are known to impact the tropospheric circulation in the subsequent weeks (Baldwin and Dunkerton, 2001). As in the case of the precursors, the surface signature of these events has recently been suggested to depend on the type of event. Some studies have shown that only S SSWs have large effects on surface climate (Mitchell et al., 2012, Seviour et al., 2013), while others have not found consistent differences between S and D SSWs in its significant surface impact (Charlton and Polvani, 2007; Cohen and Jones, 2011). Thus, there is not yet a consensus in this regard, probably due

to the differences in the algorithms used to identify S and D SSWs (Maycock and Hitchcock, 2015). As for WN1 and WN2 SSW, their surface signature has not yet been explored.

SSWs are a key element when analyzing stratospheric variability. The frequency and seasonality of SSWs are common metrics to assess the effects of tropospheric and oceanic phenomena on stratospheric variability or the stratospheric response to climate change (e.g.: Taguchi and Hartmann, 2006; Charlton-Perez et al., 2008; Ayarzagüena et al., 2018). Indeed, in modeling studies

most of them use simulations that are previously validated by comparing their results with reanalysis datasets (e.g.: Charlton et al., 2007; McLandress and Shepherd, 2009; Kim et al., 2017). Interestingly, the number of reanalyses has largely increased in the last decade, and although the assimilation data sources are the same, the reanalysis models are different, and so may the final products be (Fujiwara et al., 2017). As it happens with other atmospheric models, reanalyses also have biases and this can affect the model evaluation (Fujiwara et al., 2017).

Due to quality improvements associated with the assimilation of satellite data, modern reanalyses, such as ERA-Interim, NASA-MERRA, and NCEP-CFSR, only cover the post-satellite period since 1979. This means that the number of available reanalyses to assess the model performance in the pre-satellite era is smaller than in the second one. In addition, the amount of data to assimilate is also limited in this period. All this might produce artificial differences in results before and after the inclusion of satellite data. Gómez-Escolar et al. (2012) documented a change of some SSW features from the pre-satellite to

the post-satellite era in NCEP-NCAR and ERA-40 reanalyses. For instance, the intra-seasonal distribution and the amplitude of the SSW-associated warming showed differences between both periods, potentially due to a change in the type of the assimilated data. With the availability of the new JRA-55 reanalysis, which is the only one that applies an advanced data assimilation scheme to upper-air data during the pre-satellite era, revisiting this topic seems appropriate.



In this study, we aim to assess the performance of the most widely used reanalyses in representing SSWs. To do so, first, the main characteristics of SSWs are examined for all datasets to quantify the degree of agreement across reanalyses. Both pre- and post-satellite periods are compared to investigate whether discrepancies among reanalyses in the representation of the main SSW characteristics depend on the examined period. Secondly, we address the dynamical forcing of SSWs in all datasets,

including precursors such as blockings. Finally, the surface impact of SSWs retrieved from the different reanalyses is analyzed. Special emphasis is made on the assessment and robustness of the potential differences in the forcing and surface impact of WN1 and WN2 SSWs.

Our work is a contribution to the Chapter 6 of the Stratosphere-troposphere Processes And their Role in Climate (SPARC) Reanalysis Intercomparison Project (S-RIP) initiative, which aims to assess stratosphere-troposphere coupling in reanalyses.

In the framework of this initiative, a few recent studies have addressed some aspects of the representation of polar stratospheric variability in reanalyses. In particular, Martineau et al. (2018) and Hitchcock (2019) also investigate SSWs-related aspects. The former analyzes the SSWs the momentum budget restricted to the post-satellite period, while Hitchcock (2019) compares the representation of stratosphere-troposphere coupling in both pre and post-satellite period, with the emphasis on the impact of including pre-1979 data. Different from these studies, our work provides a comprehensive inter-reanalyses comparison of

the most important and typical aspects and processes associated with SSWs in both pre- and post-satellite era. Additionally, we explore further the characteristics of WN1 and WN2 SSWs that have not been investigated yet.

The structure of the paper is as follows. The data used and methodology applied are described in Section 2. Section 3 compares the performance of the main characteristics of SSWs across reanalyses. Section 4 focuses on the dynamical forcing of the events and Section 5 addresses the performance of reanalyses in representing the surface impact of SSWs. The main

conclusions are summarized in Section 6.

## 2 Data and methodology

### 2.1 Data

We have used daily data from the following reanalyses: ERA-40 (Uppala et al., 2005), ERA-Interim (Dee et al., 2011), JRA-25 (Onogi et al., 2007), JRA-55 (Kobayashi et al., 2015), NASA-MERRA (Rienecker et al., 2011), NCEP-CFSR (Saha et al.,

2010), NCEP-DOE (Kanamitsu et al., 2002), and NCEP-NCAR reanalysis (Kalnay et al., 1996). More details about the different reanalyses can be found in Fujiwara et al. (2017). For the comparison across reanalysis, all data have been interpolated to a common regular grid of 2.5° lon x 2.5° lat.

The methodology for the intercomparison follows the S-RIP specifications. As such, the analysis has been carried out for two different periods: historical (1958-1978) and comparison (1979-2012). Given the periods covered by each reanalysis, only

ERA-40, NCEP-NCAR, and JRA-55 are employed in the historical period. In contrast, all the above listed reanalyses are used in the comparison period with the exception of ERA-40, because it ends in 2002. The performance of each reanalysis is evaluated against a multi-reanalysis mean (MRM), herein considered as an "unbiased" reference. In the historical period, the



MRM refers to the average of the three reanalyses that cover that period, while in the comparison period, the MRM is defined as the average of the most recent reanalyses of each center (ERA-Interim, NCEP-CFSR, JRA-55 and NASA-MERRA). Hereafter, anomalies for each reanalysis are defined as the departure of the field from the daily climatology. In the historical period, the climatology covers the whole period (i.e. 1958-1978), whereas the comparison period uses the 1981-2010 baseline.

Unless otherwise stated, statistical significance of the results is computed with a Monte-Carlo test of 1000 permutations, each one containing the same number of cases and dates as the SSWs of each composite but with random years of occurrence.

### 2.2 Criteria for the identification of SSWs

Unless otherwise stated, we have used the list of SSWs and common dates identified in Butler et al. (2017) and provided for the S-RIP initiative (Chapter 6). First, for each reanalysis, SSWs are identified based on the reversal of the zonal mean zonal

wind at 60ºN and 10hPa between November and March, with at least 20 days of separation between events. Stratospheric final warmings are excluded by imposing at least 10 consecutive days of westerly winds before the end of April. The first day of reversal of winds determines the date of occurrence of the SSW (the so-called central date). Common SSWs are those identified by at least two of the three reanalyses in the historical period and by at least four out of seven reanalyses in the comparison period. The central date of these common events is computed as the average of the central dates from the SSWs detected for

each reanalysis. Thus, with this approach, the same events and central dates apply for all reanalyses even if the reversal of the winds does not occur in all of them. This is useful to ensure that the differences between datasets are not due to the selection of different events or dates. The common SSWs are listed in Table 1 for the comparison period.

Nevertheless, in the very first part of our study, we have addressed the opposite question and quantified the possible discrepancies in the frequency of SSWs among reanalyses when the same criterion is applied to all datasets. In that case, we

have imposed the WMO definition for the identification of SSWs in each reanalysis. The definition is based on the simultaneous reversal of zonal-mean zonal wind at 10hPa and 60ºN and zonal-mean temperature difference between 90ºN and 60ºN at the same level (Labitzke, 1981).

### 2.3 Types of SSWs

SSWs are classified following two definitions: D vs S SSWs, and WN1 vs WN2 events. In this study, D and S SSWs were

identified according to the algorithm by K. Shibata (personal communication), which is similar to that of Charlton and Polvani (2007). It is based on the identification of cyclonic vortices and their relative sizes by means of the absolute vorticity at 10hPa from 5 days before to 10 days after (i.e. [-5,10]-day) with respect to the occurrence of an SSW. The events were classified individually in each reanalysis. The classification into S/D events of common SSWs (used in Sections 4 and 5) was based on the predominant type across reanalyses, similarly to the identification of the common dates (Table 1).

WN1 and WN2 SSWs were selected by applying a zonal Fourier decomposition of the daily 50hPa geopotential height data at 60ºN into WN1 ($Z_1$) and WN2 ($Z_2$) amplitudes for the [-10,0]-day period before each SSW (Barriopedro and Calvo, 2014). An SSW was defined as a WN2 event if $[Z_2] \geq [Z_1]$ (brackets denote the averaged amplitude for the [-10,0]-day period before the



SSW) or if $Z_2 - Z_1 \geq 200$ m at least for one day within the [-10,0]-day period before the SSW. Otherwise, the SSW was defined as a WN1 event. See the list of events of each type in Table 1.

## 2.4 Dynamical benchmarks

We have applied the following diagnostics proposed by Charlton and Polvani (2007) to evaluate the dynamical signatures associated with the occurrence and development of SSWs:

- Amplitude of the SSW in the middle stratosphere (hereafter *amp010*) computed as the area-weighted mean 10hPa temperature anomaly over the polar cap (50°N-90°N) and averaged for the [-5,5]-day period with respect to the central date of the event.
- Amplitude of the SSW in the lower stratosphere (hereafter *amp100*), defined as *amp010* but at 100hPa. It provides a measure of the coupling between the middle and lower stratosphere around the occurrence of SSWs.
- Deceleration of the polar night jet (PNJ) (hereafter *decelu*), corresponding to the difference of the 10hPa zonal-mean zonal wind at 60°N between the [-15, -5]-day period prior to the central date and the [0, 5]-day period after the central date.
- Wave activity prior to SSW (hereafter *actwav*), computed as the area-weighted mean 100hPa meridional eddy heat flux (HF) anomaly averaged over 45°N-75°N for the [-20,0]-day period before the occurrence of the event.

## 2.5 Upward-propagating wave activity

The anomalous meridional eddy HF averaged over 45°N-75°N at different pressure levels was used as a metric to measure the upward vertical propagation of wave activity. This latitudinal band corresponds to the climatological area with the strongest vertical wave propagation from the troposphere (Hu and Tung, 2003).

As a second step, the methodology by Nishii et al. (2009) was applied to analyze the role of different forcing processes in the occurrence of SSWs. This methodology is based on the decomposition of daily anomalous eddy HF into two components, which correspond to the interaction between climatological waves and anomalous waves (second and third right hand terms of Eq. 1) and the inherent contribution of anomalous waves (first right hand term of Eq. 1):

$$[v^*T^*]_a = [v_a^*T_a^*]_a + [v_c^*T_a^*] + [v_a^*T_c^*] \tag{1}$$

where brackets and asterisks indicate zonal mean and deviation from it, respectively, $v$ is meridional wind, $T$ is temperature and the $a$ and $c$ subscripts denote anomalies and climatological values, respectively. Eq. 1 has been applied to each pressure level.

## 2.6 Blocking definitions

The precursor role of blocking in SSWs has been discussed with discrepancies across studies (see e.g., Castanheira and Barriopedro (2010) for an overview on this topic). The divergent results of previous studies may partially be attributed to different methodologies of blocking detection (e.g., Woollings et al., 2008). In this study, three different blocking definitions



have been used to address this question. The three methodologies use daily geopotential height at 500 hPa (Z500) and span almost all approaches to blocking definition. The first method is based on the occurrence of regional and persistent meridional Z500 gradient reversals (the absolute method, ABS; e.g., Scherrer et al., 2006). The second metric involves the detection of persistent and quasi-stationary Z500 anomalies, computed with respect to the local climatological field (the anomaly method, ANO; e.g., Sausen et al., 1995). Finally, a combined method of absolute and anomaly Z500 fields (the mixed method, MIX) is used, providing a two-folded perspective of blocking (Barriopedro et al., 2010). Several criteria are imposed to ensure that the detected episodes represent large-scale, quasi-stationary, and persistent high-pressure systems. See Woollings et al. (2018) for more details about blocking definitions.

## 3 Main SSW characteristics

In this section, the main signatures of SSWs (frequency, type of events and process-based diagnostics) are analyzed for each period and compared among the different datasets.

### 3.1 Frequency, seasonality and type of events

First, we have analyzed the discrepancies in the frequency and type of events across reanalyses when the same criterion is applied to each dataset. Table 2 shows the mean frequency of events and the ratio of D to S SSWs for each period and reanalysis. The main differences are found in the historical period when the reanalyses show a large spread in both frequency and type of events. In particular, the NCEP/NCAR reanalysis displays the results that deviate the most from the other two datasets, although the differences are not statistically significant at the 95% confidence level (Student's t-test). The short period of analysis and hence the reduced sample might explain part of these discrepancies. More importantly, the unavailability of satellite data in the pre-satellite era leads to a strong dependency of the reanalysis data in the stratosphere on the characteristics of each reanalysis model. Note that NCEP/NCAR reanalysis is the only reanalysis with a low-top model and a lid in the stratosphere (3hPa), whereas JRA-55 and ERA-40 have the top in the mesosphere (0.1hPa). The low top typically dampens variability close to the top and so, reduces the probability of the occurrence of an SSW (Charlton-Pérez et al., 2013). In fact, the standard deviation of daily polar temperature and zonal wind at 10 hPa in December and January is much lower in NCEP/NCAR than in the other reanalyses, although the differences are not statistically significant at the 95% confidence level (F-Fisher test) (Figure 1a, c). In contrast, at lower levels, we do not find such discrepancies (see 100 hPa temperature in Fig. 1b, d), supporting that the occurrence of SSWs during this period is strongly influenced by the model performance and hence should be considered reanalysis-dependent.

Conversely, in the comparison period, there is a good agreement in both the frequency and ratio of D/S SSWs. Small differences are found, particularly, in the D/S ratio, but this might be due to the specific thresholds or other methodological



issues of the applied criterion, since such deviation does not appear when classifying SSWs into WN1 and WN2 events (Barriopedro and Calvo, 2014).

Regarding SSWs seasonality, Figure 2 shows the SSW total frequency distribution within ±10-day periods. Similarly to the winter mean frequency of SSWs, historical reanalyses show the largest spread in the seasonal distribution. In particular, ERA-40 and JRA-55 display increasing SSW occurrence from early winter that maximizes in January and decreases by late winter (Fig. 2a), in agreement with the temporal evolution of the standard deviation of the zonal-mean zonal wind at 60ºN and 10 hPa (Fig. 1c). In contrast, SSWs for NCEP/NCAR are more uniformly distributed with three sharp maxima in early, mid and late winter. The early winter peak of SSWs in NCEP/NCAR can be traced back to the PNJ, which shows weaker values than the other two reanalyses. This difference in the PNJ is not statistically significant, though, likely due to the short sample and the general large interannual variability of the winter polar stratosphere (Fig. 1c). On the other hand, the lower wind variability in January in NCEP/NCAR would agree with the reduced frequency of SSWs in that month and reanalysis, as compared to the other datasets. In the comparison period the results are similar across reanalyses (Fig. 2b). In this period, the maximum occurrence shifts to late winter in all datasets compared to the distributions of ERA-40 and JRA-55 in the historical period. Similar differences in the intra-seasonal distribution of events were already documented by Gómez-Escolar et al. (2012) between the pre- and post- 1979 periods. This adds support to the hypothesis of decadal variability in the intra-seasonal occurrence of SSWs.

## 3.2 Process-based diagnostics

The processes involved in the occurrence of SSWs have been compared across reanalyses by using the diagnostics defined in Section 2d. In this case, and in the rest of the paper, we have used the common dates of SSWs to make sure the differences found across reanalyses are not due to the inclusion of different events.

Figure 3 shows the statistics (mean, median and interquartile range) of the dynamical benchmarks for all reanalyses in the two periods. A quick comparison of the MRM of these benchmarks for both periods reveals that SSWs are preceded by a similar anomalous strengthening of wave activity at 100hPa, are associated with a comparable deceleration of the PNJ and have a similar amplitude in the middle and lower stratosphere in both periods. Only slight differences are found in the median of *decelu* and *amp100* (compare Fig. 3b,c with Fig.3e,f). However, given that the median and mean of these magnitudes for one period are included within the interquartile range of the other, we can conclude that SSWs characteristics are similar in both periods of study.

The comparison period shows good agreement among all reanalyses as all datasets are characterized by similar median, mean and spread values (Fig. 3e-h). Nevertheless, slight deviations can be found for NCEP/NCAR in the distribution of *decelu*, which is shifted towards lower values and shows a reduced spread among events, as compared to the rest of the datasets (Fig. 3g). These deficiencies are even clearer in the historical period, when a similar discrepancy is detected in *amp010* (Fig. 3a), consistent with the reduced strength and variability of the PNJ in NCEP/NCAR reanalysis (Figs. 1c). As the deviation of *decelu* in the NCEP-NCAR reanalysis is common for both periods, this might point to a bias of the model, whose effects are amplified





in the first period by the lower amount of assimilated data. As mentioned before, this bias is very likely linked to the low top of the model, provided that the SSW characteristics at lower levels (i.e. *amp100*, *actwav*) do not differ much from those of other reanalyses. Note that these differences are still noticeable in NCEP-DOE but they are minimized, particularly for *decelu*, arguably due to improvements introduced in the new version of this reanalysis such as a new ozone climatology (Kanamitsu et al., 2002).

A similar analysis has been carried out separately for WN1 and WN2 SSWs in the comparison period (Fig. S1). All datasets reproduce a similar behavior for both types of events and all diagnostics, with the exception of the associated deceleration of the PNJ in the middle stratosphere: WN2 SSWs are related to larger decelerations of the PNJ, probably because they are usually preceded by a stronger polar vortex than WN1 ones (Albers and Birner, 2014; Díaz-Durán et al., 2017). These results also confirm the overall good agreement across reanalyses except for the deficiency of NCEP/NCAR concerning *decelu*. Unfortunately, these findings cannot be confirmed in the historical reanalyses due to the very low frequency of WN2 events in that period (not shown).

## 4 Dynamical forcing

### 4.1 Upward-propagating wave activity

Figures 4 and 5 show the composited anomalous eddy HF, averaged between 45° N and 75°N, at different levels around the SSWs onset date for the historical and comparison period, respectively. Only results from 100 to 10 hPa are presented, as these are the levels with the strongest HF anomalies. The MRM shows a strong anomalous peak of HF around the central date of SSWs in both periods. This strong peak is preceded by a weak pulse around [-20, -15] days in the middle stratosphere in the comparison period but not in the historical one. The largest reanalyses deviation is detected in the middle stratosphere in agreement with Martineau et al. (2018), and they are more pronounced for the historical than for the comparison period.

By applying the methodology by Nishii et al. (2009) we have analyzed the contributing role of the different HF terms to the occurrence of SSWs. The MRM decomposition of the HF in the comparison period shows that the strongest peak ([-5,0]-day interval) is mainly due to the action of anomalous waves, albeit with a relevant contribution of the constructive interaction between climatological and anomalous waves (Figs. 4c, e and 5c, e). Conversely, the precedent weaker pulses of the comparison period seem to be more dominated by the interaction term. The agreement among reanalyses concerning the relative roles of these terms is higher for the comparison period, mainly in the middle stratosphere, than for the historical one (compare Fig. 4d, f vs Fig. 5d, f).

Given the documented differences in the dynamical forcing of different types of SSWs (e.g. Smith and Kushner, 2012; Barriopedro and Calvo, 2014), we have repeated the analysis separately for WN1 and WN2 SSWs (Fig. 6). It has only been done for the comparison period, due to the low sample size of WN2 events for the historical one. Although there is not a univocal relationship between D and S SSWs and WN1 and WN2 events (Waugh, 1997), our results for WN1 and WN2 events agree well with those of Smith and Kushner (2012) for D and S SSWs. WN1 events are mainly triggered by persistent but



moderately intense anomalies of HF during different periods ([-20, -15] and [-10, 0] days), which are associated with the constructive interference of climatological and anomalous waves (Figs. 6e and i). In contrast, WN2 events are related to intense but short pulses of eddy HF in the five days prior to the central date. These pulses are predominantly due to the anomalous term (Figs. 6g and k), consistent with Smith and Kushner's finding for S SSWs. The recovery of the polar vortex after WN2 SSWs is due to a reduction of wave activity in the interaction term, while only the anomalous term has a statistically significant contribution to this reduction after WN1 SSWs (Figs. 6e, g, i and k).

The comparison among reanalyses reveals that all datasets can reproduce the above differences between WN1 and WN2 SSWs. The spread is higher for WN2 SSWs than for WN1 SSWs (Figs. 6b, d, f, h, j, and l), particularly for the anomalous HF term (Fig. 6l). However, considering the differences in HF values between WN1 and WN2 SSWs (i.e., by dividing the standard deviation by the MRM), the resulting spread becomes comparable for both types of SSWs (not shown).

### 4.2 Tropospheric circulation anomalies associated with SSWs

To investigate the tropospheric patterns preceding SSWs we have analyzed the averaged Z500 anomalies in the 10 days prior to the central date of each type of SSW (Fig. 7). As in the previous section, we have focused on the differences between WN1 and WN2 events in the comparison period only. The chosen time window corresponds to the peak of the strongest anomalies of HF in Fig. 5a. The results reveal statistically significant differences between the precursors of WN1 and WN2 SSWs (Fig. 7c). The precursor signal for WN1 SSWs shows a predominant WN1-like structure, with negative anomalies of Z500 over the North Pacific and eastern Asia, and positive anomalies over northern Canada, the North Atlantic and western Siberia (Fig.7a). This agrees with the pattern identified by previous studies such as Limpasuvan et al (2004) and Garfinkel et al. (2012) for all SSWs. Most of these centers of action project onto the climatological WN1 of the MRM, especially the one over the North Pacific (e.g., Garfinkel and Hartmann, 2008), explaining the high positive values of the interaction term of HF (e.g., Martius et al., 2009; Nishii et al., 2011). Differently, the precursor signal of WN2 SSWs shows strong negative Z500 anomalies over Canada and Greenland and positive anomalies over the northeastern Pacific (Fig. 7d). The main anomalous centers coincide geographically and in sign with the antinodes of the climatological WN2 of the MMR (e.g., Garfinkel and Hartmann, 2008). Although this pattern agrees with the preferred blocking precursors of WN2 SSWs (Barriopedro and Calvo, 2014), it seems counterintuitive with the predominant role of the anomalous waves found in Fig. 6 for these events, although we are looking at very different levels in the two figures. The same apparent contradiction was already highlighted by Smith and Kushner (2012). However, additional analyses revealed that, despite the projection of Z500 anomalies onto the stationary WN2, the interaction HF term is weak due to the low amplitude of WN2 climatological $v^*$ and $T^*$ with respect to that of anomalous WN2 waves. Consequently, a considerable part of the WN2 HF anomalies is explained by the large amplitude of the anomalous WN2 wave preceding these events (not shown).

The agreement among reanalyses is very good (Fig.7b and e). Only very small differences appear in the tropospheric pattern over the North Pacific, which are larger for WN2 than for WN1 SSWs, in agreement with the comparison of wave activity (Fig.6). We stress that the largest differences in wave activity among reanalyses are found in the middle stratosphere and hence





the Z500 deviations from the MRM are smaller than those in the HF composites. The lower spread among reanalyses in tropospheric fields compared to that in the stratosphere is expected based on the larger number of assimilated data.

## 4.3 Blocking

The positive Z500 anomalies identified in the previous section may imply an increased blocking frequency over those locations prior to the occurrence of each type of SSW. Similarly, a below-normal activity of blocking before SSWs might translate into negative Z500 anomalies. Here, we identify blocking precursors of WN1 and WN2 SSWs by performing 2-D composites of the blocking frequency for the [-10,0]-day period before the central day of SSWs (same window as in Fig. 7). We have employed the three different algorithms described in Section 2f. Upper and middle rows of Figure 8 show the MRM of blocking precursor frequencies for WN1 and WN2 SSWs in the comparison period, respectively. Bottom row of Figure 8 displays the MRM of the mean blocking frequency prior to all SSWs (a pseudo-climatology). In general, in all methods there is a spatial preference for specific blocking precursors depending on the main wave activity preceding SSWs. For WN1 SSWs, enhanced (above climatology) blocking frequencies are detected over the western Atlantic and east of Scandinavia, and reduced (below climatology) blocking activity occurs over the eastern Pacific (compare upper and bottom rows of Figure 8). Nearly opposite patterns are identified for WN2 SSWs (compare middle and bottom rows of Figure 8) except for an increased blocking frequency over east of Scandinavia. These results also agree well with the Z500 pattern preceding each type of SSWs in Fig. 7. They are also consistent with previous studies that identified the preferred location of blockings for the intensification of WN1 and WN2 wave activity (e.g., Castanheira and Barriopedro, 2010; Nishii et al., 2011; Barriopedro and Calvo, 2014; Ayarzagüena et al., 2015).

This blocking signal is reproduced by all methods and reanalyses (not shown), although the intensity, significance and spatial extension of the anomalies vary with the blocking definition. For example, the precursor signal of SSWs in ABS is confined to smaller regions than in ANO and MIX, eventually becoming non-significant. These differences between methods do not only refer to the blocking signal prior to SSWs but also to the climatology (Figs. 8g-i), which can be explained by the different aspects captured by each blocking indicator (Barriopedro et al. 2010). Reanalyses show a reasonable agreement in the blocking frequency results, and they even agree on the statistical significance of changes in the blocking frequency for the ANO and MIX methods, which show a noticeable deviation from the climatology prior to SSWs. Thus, the disagreement between previous studies regarding the precursor role of blocking in SSWs is better explained by the blocking definition than the chosen reanalysis.

## 5 Surface signal of SSWs

Finally, the surface signal after the occurrence of SSWs was explored by compositing the mean sea-level pressure (MSLP) anomalies of the [5, 35]-day period for all events. The time interval was selected following Palmeiro et al. (2015), who identified the strongest negative values of the Northern Annular Mode (NAM) index in this period. We found a general good





agreement in the surface signal of all SSWs across reanalyses in both historical and comparison periods (not shown). Similar to the previous sections, we present here only the MSLP composites for WN1 and WN2 SSWs and the comparison period (Figs. 9a and d). WN1 and WN2 SSWs show a significant negative NAM-like pattern response, with positive anomalies over the polar cap in both cases. However, some slight differences between WN1 and WN2 events are found. Over the northeastern

Pacific, MSLP anomalies of different sign (positive for WN2 SSWs and negative for WN1 SSWs) were also detected prior to the occurrence of SSWs (see Fig. 7 and also in MSLP maps (not shown)). Thus, they may be a remainder of the tropospheric precursors, as also suggested by Charlton and Polvani (2007). In the Euro-Atlantic sector, negative anomalies after WN1 SSWs extend over the whole Atlantic Ocean and western and central Europe (Fig. 9a), while those related to WN2 SSWs are shifted towards Eurasia (Fig. 9d). Nevertheless, these differences are only statistically significant in western-central Europe and the

Mediterranean region, where the response to SSWs is significantly stronger in WN2 than in WN1 SSWs (Fig. 9c). Interestingly, despite their small extension, the different surface responses for WN1 and WN2 SSWs reported here show very good agreement across reanalyses (Figs. 9b and e). Note that the deviations from the MRM are very low for both types of SSWs. Additionally, the regions with the highest disagreement across reanalyses do not correspond to the areas with the largest differences in the surface fingerprint of WN1 and WN2 SSWs. Thus, although small, the differences in surface responses

detected between both types of events are robust across reanalyses.

In the last decades, many studies have focused on the surface signal of D and S SSWs (e.g.: Charlton and Polvani, 2007; Mitchell et al., 2013; Lehtonen and Karpechko, 2016). However, this classification is difficult to predict before the SSW onset, since it is strongly based on the evolution of the polar vortex during the post-warming phase. Here, we have rather investigated the surface signal of WN1 and WN2 SSWs, whose typification is dictated by their precursors. Indeed, whereas the Z500

patterns preceding SSWs show statistically significant differences for WN1 and WN2 events (Fig. 7c), the areas with statistical significance of the differences between D and S events is more limited (Fig. 7f). In the case of the surface signal, both classifications (WN1/WN2 or S/D) show areas of statistically significant differences between the two types of events, being stronger for WN1/WN2 than for D/S SSWs (compare Figs. 9c and 9f). Our results agree well with previous studies that also found a surface signal for D and S SSWs (e.g., Charlton and Polvani, 2007; Maycock and Hitchcock, 2015). Maycock and

Hitchcock (2015) indicated that the absence of a surface fingerprint for D SSWs reported by previous studies is more probably due to the sampling of events rather than a physical reason. The reported differences between the surface impacts of WN1 and WN2 SSWs may also be influenced by this issue, particularly taking into account the small sampling size of WN2 events. Still, our results confirm a detectable surface fingerprint for all types of SSWs independently of the classification chosen.

## 6 Summary and conclusions

In this study, we have compared the representation of the main features, triggering processes and surface fingerprint of SSWs in different generations of reanalyses. Apart from a direct assessment of the SSW characteristics in the pre- and post-satellite period, questions concerning the representation of SSWs by reanalyses have been addressed thanks to the larger number of



datasets available for the post-1979 period. Unlike most studies that focus on D versus S SSWs, a separate analysis of WN1 and WN2 events has also been performed. The main conclusions are summarized as follows:

- An overall good agreement across reanalyses is found in the representation of the main features of SSWs. However, there are differences across reanalyses, particularly in the historical period, concerning the characteristics of SSWs in the middle stratosphere such as amplitude or deceleration of the PNJ. Some of the discrepancies also extend to climatological fields and their variability and are more pronounced for the NCEP/NCAR reanalysis. Arguably, the characteristics of the reanalysis models, including the location of their upper lid, play an important role in that period, when the performance of the reanalysis is preferentially determined by the characteristics of the underlying model. These limitations also affect the comparison period, but at much less extent, due to the availability of satellite data in the upper levels.

- In general, SSWs (frequency, type and dynamical benchmarks) do not substantially differ between the historical and comparison periods. Only the seasonal distribution of SSWs reveals robust differences between both periods with a shift towards a later occurrence in the satellite period, in agreement with Gómez-Escolar et al. (2012) and Hitchcock (2019).

- SSWs are mainly associated with anomalous wave packets immediately before their onset. However, the interference with climatological stationary waves plays a predominant role several days before the SSW onset. This behavior is robust across reanalyses during the comparison period, but subject to considerable uncertainties during the historical period concerning the wave activity in the middle stratosphere.

- WN1 and WN2 SSWs and their tropospheric precursors display differences in the comparison period that are robustly captured by all reanalyses. WN1 events are mainly triggered by the interaction between climatological and anomalous waves during long-lasting and moderately intense peaks of HF anomalies. Conversely, WN2 events are related to intense but short-lived pulses of HF arising from anomalous wave packets. The results resemble those by Smith and Kushner (2012) for D and S events, respectively, despite the lack of a one-to-one correspondence between WN1 (WN2) and D (S) SSWs.

- The tropospheric precursor signal for WN1 and WN2 SSWs shows a predominant WN1-like and WN2-like structure, respectively. This is consistent with the spatial distribution of blockings preceding both types of SSWs. For WN1 SSWs, there is an enhanced activity over the western Atlantic and below normal frequencies over the eastern Pacific, with nearly opposite patterns for WN2 SSWs. A robust pattern emerges for all reanalyses, but there are substantial differences among blocking definitions.

- Both WN1 and WN2 SSWs have significant impacts on surface weather characterized by a negative NAM pattern, but with some differences in southern and central Europe. These differences are significantly different between WN1 and WN2 events and robust across reanalyses during the comparison period.

In summary, we conclude that the representation of SSWs is, in general, robust in both periods of study for the available reanalyses, and overall not different between the pre- and post-satellite eras. This would agree with Hitchcock (2019) who

recommended the consideration of using data prior to 1979 in dynamical studies for stratosphere-troposphere coupling, as it might be advantageous for reducing the sampling uncertainty for many purposes. However, in our study some discrepancies in the historical period were identified, particularly for the NCEP/NCAR reanalysis, which limit its use for this period in model evaluation initiatives. Furthermore, this work provides some guidelines, highlighting discrepancies among reanalyses concerning SSWs and identifying related aspects that may be sensitive to the chosen reanalysis. Although robust, some reanalyses results (such as the differences between types of SSWs) should be taken with caution in this period, due to the limited sampling.

**Author contribution**

BA, FMP, DB, NC and UL designed the analysis and wrote the paper. BA, FMP and DB carried out the analysis of the renalyses data and drafted the figures. KS provided the algorithm for identification of S and D SSWs and helped with its implementation.

**Data availability**

NCEP/NCAR and NCEP-DOE reanalyses data were provided by the NOAA/OAR/ESRL PSD, Boulder, Colorado, USA, from their Web site at http://www.esrl.noaa.gov/psd/. JRA-25 data were provided from the cooperative research project of the JRA-25 long-term reanalysis by the Japan Meteorological Agency (JMA) and the Central Research Institute of Electric Power Industry (CRIEPI). Japanese 55-year Reanalysis (JRA-55) project was carried out by the Japan Meteorological Agency (JMA). JRA-25, JRA-55 and NCEP-CFSR data were accessed through NCAR/UCAR Research Data Archive (https://rda.ucar.edu). The NASA-MERRA data were disseminated by the Global Modeling and Assimilation Office (GMAO) and the GES DISC. ERA-Interim and ERA-40 are available online (http://apps.ecmwf.int/datasets/data/).

**Acknowledgments**

BA was funded by "Ayudas para la contratación de personal postdoctoral de formación en docencia e investigación en los departamentos de la UCM". This work was supported by the European Project 603557-STRATOCLIM under program FP7-ENV.2013.6.1-2 and Spanish Ministry of Economy and Competitiveness through the PALEOSTRAT (CGL2015-69699-R) project.

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



**Table 1: Classification of the common SSWs into WN1 and WN2 events in the comparison period. (In brackets the S/D classification).**

| WN1 SSWs | | WN2 SSWs |
|---|---|---|
| 29 02 1980  (D) | 11 02 2001 (D) | 22 02 1979 (S) |
| 04 03 1981  (D) | 31 12 2001 (D) | 01 01 1985 (S) |
| 04 12 1981  (D) | 18 01 2003 (S) | 21 02 1989 (S) |
| 24 02 1984  (D) | 05 01 2004 (D) | 20 03 2000 (D) |
| 23 01 1987  (D) | 21 01 2006 (D) | 22 02 2008 (D) |
| 08 12 1987  (S) | 24 02 2007 (D) | 24 01 2009 (S) |
| 14 03 1988  (S) | 09 02 2010 (S) | |
| 15 12 1998  (S) | 24 03 2010 (D) | |
| 26 02 1999  (S) | | |

**Table 2: Frequency of SSWs per decade and ratio of vortex displacement (D) vs vortex split (S) SSWs for each reanalysis and period of study.**

| Reanalyses | Historical period (1958-1978) | | Comparison period (1979-2012) | |
|---|---|---|---|---|
| | Frequency (SSWs/dec) | Ratio D/S | Frequency (SSWs/dec) | Ratio D/S |
| ERA-40 | 6.2 | 1.6 | | |
| NCEP-NCAR | 4.3 | 0.5 | 6.2 | 1.6 |
| JRA-55 | 5.2 | 0.8 | 6.8 | 1.2 |
| ERA-Interim | | | 6.2 | 1.6 |
| JRA-25 | | | 6.5 | 1.8 |
| NCEP-CFSR | | | 6.5 | 1.4 |
| NCEP-DOE | | | 6.5 | 1.4 |
| NASA-MERRA | | | 6.5 | 1.2 |





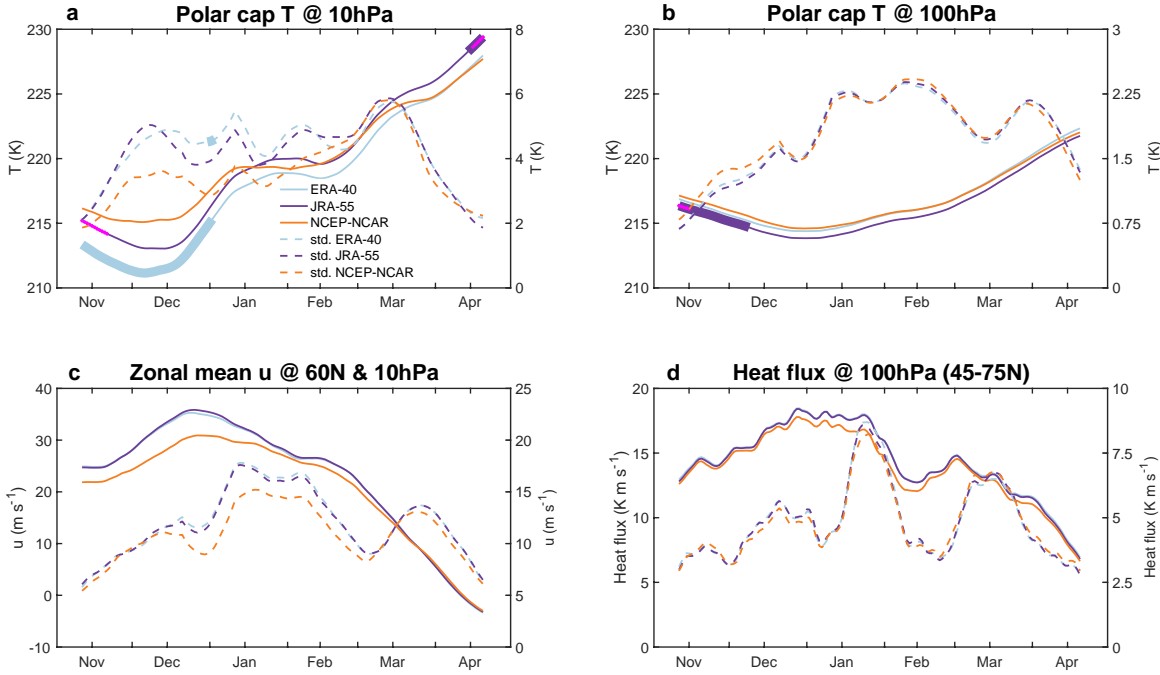

**Figure 1: 21-day running mean of the daily climatology (solid line) and standard deviation (dashed line) in the historical period (1958-1978) of: (a) polar-cap (50ºN -90ºN) averaged temperature at 10 hPa, (b) polar-cap (50ºN -90ºN) averaged temperature at 100 hPa, (c) zonal mean zonal wind at 60°N and 10 hPa and (d) heat flux at 100 hPa averaged over 45ºN -75°N. The left (right) y-axis refers to the mean (standard deviation) in each plot. Thick lines indicate values of ERA-40 or JRA-55 that are significantly different from those of NCEP-NCAR reanalysis at the 95% confidence level. Magenta crosses correspond to JRA-55 values that are significantly different from ERA-40 ones at the 95% confidence level (Student's t-test).**





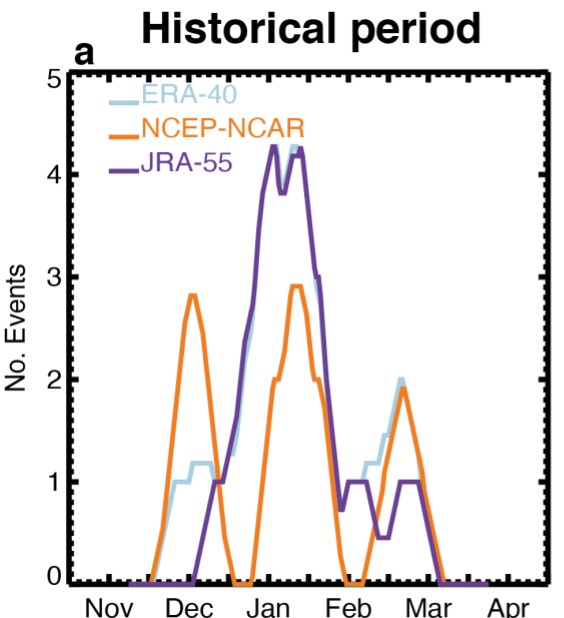
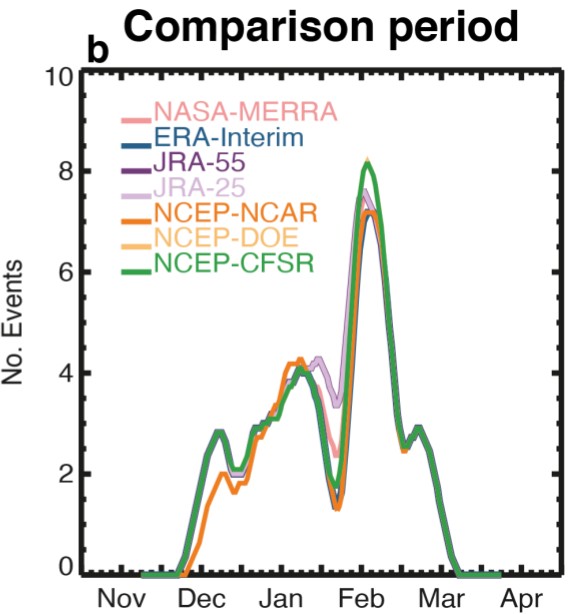

**Figure 2.** SSW total frequency distribution within ±10 day periods from the date displayed in the x-axis for: (a) the historical period (1958-1978) and (b) the comparison period (1979-2012).





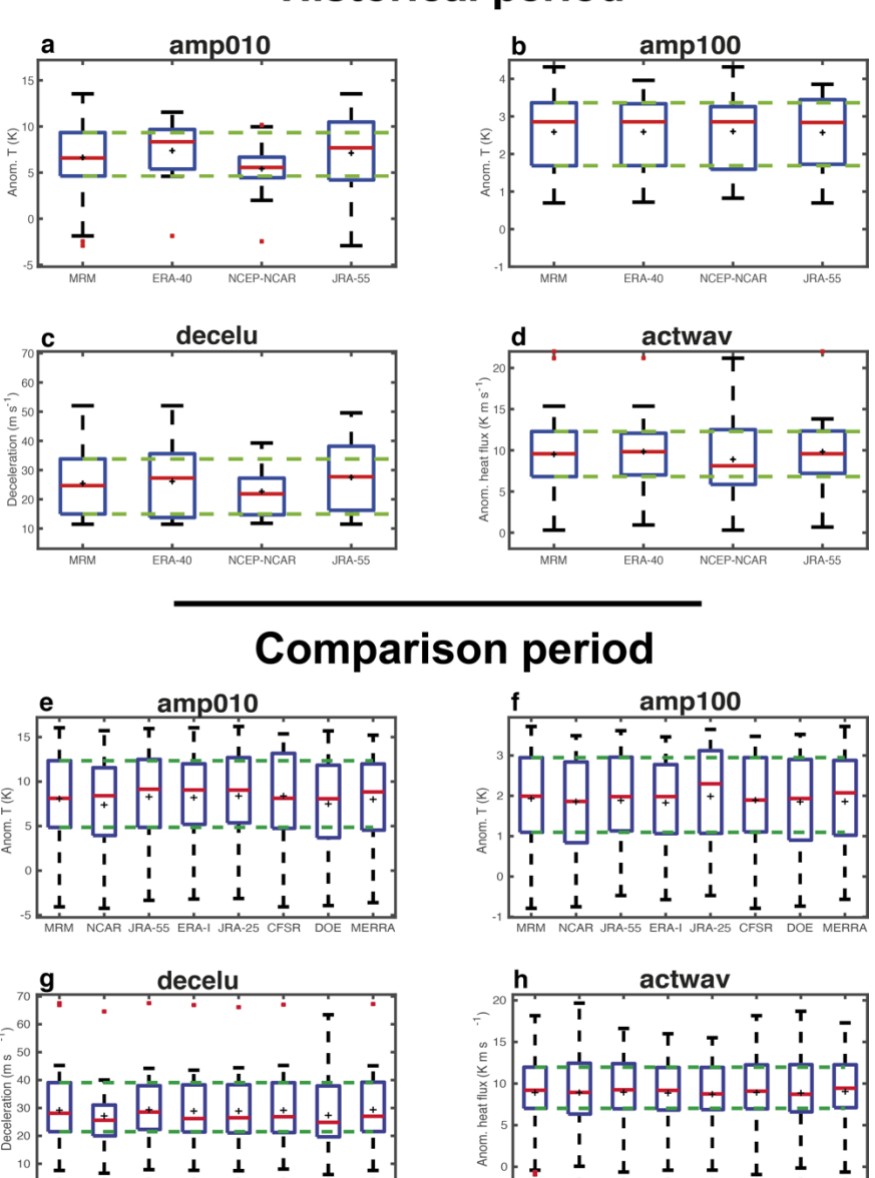

**Figure 3.** Box plots showing the distribution of the dynamical benchmarks of SSWs (amp010, amp100, decelu and actwav) in the historical (1958-1978) and comparison (1979-2012) periods. The interquartile range is represented by the size of the box and the red line (black cross) corresponds to the median (mean). Whiskers indicate the maximum and minimum points in the distribution that are not outliers. Outliers (red crosses) are defined as points with values greater than 3/2 times the interquartile range from the ends of the box. See text for the definition of dynamical benchmarks.





**Figure 4. (a)** Composited time evolution of the total anomalous heat flux averaged over 45°N-75°N (K m s⁻¹) at different pressure levels from 29 days before to 30 days after the occurrence of SSWs in the historical (1958-1978) period. Contour interval is 20 K m s⁻¹. **(b)** Same as (a) but for the standard deviation of the reanalyses with respect to the MRM divided by the square root of the number of reanalyses. Contour interval is 1 K m s⁻¹. **(c) and (d)** Same as (a) and (b) but for the interaction between climatological and anomalous waves. Contour intervals are 10 K m s⁻¹ and 2 K m s⁻¹, respectively. **(e) and (f)** Same as (a) and (b) but for the contribution of the anomalous waves to the total anomalous heat flux. Contour intervals are 10 K m s⁻¹ and 2 K m s⁻¹, respectively. Shading in (a), (c) and (e) denotes statistically significant anomalies at the 95% confidence level of the same sign in at least 66.7% of all reanalyses (Monte-Carlo test).



**Figure 5.** Same as Fig. 4 but for the comparison (1979-2012) period.



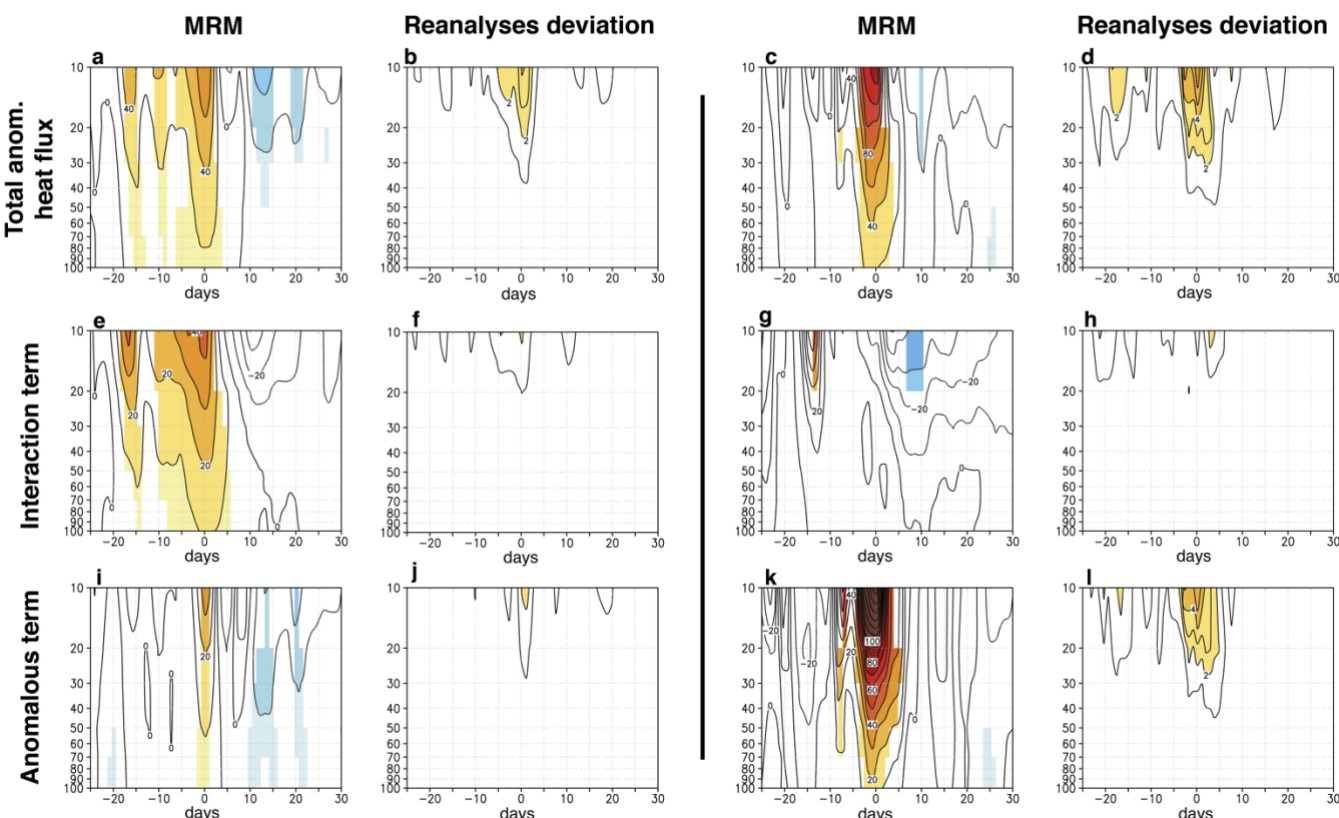

**Figure 6.** Same as Fig. 5 but for WN1 SSWs (left) and WN2 SSWs (right).





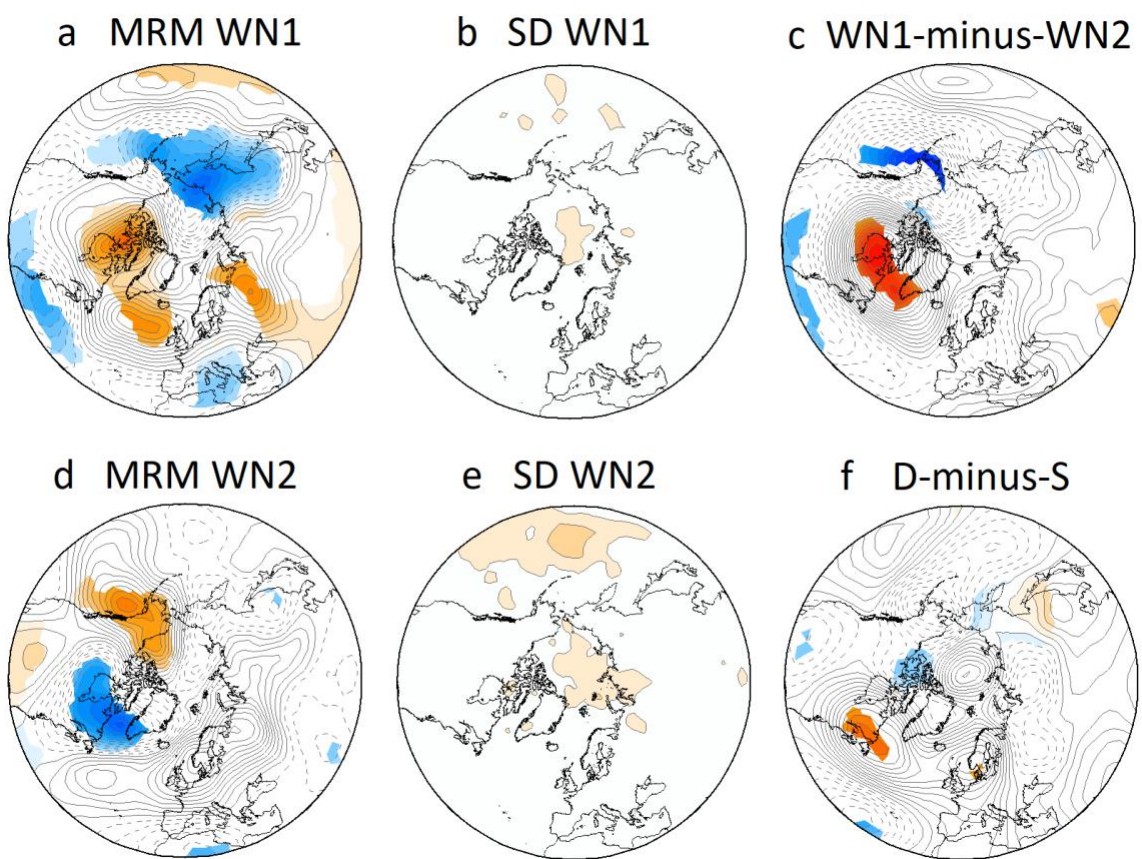

**Figure 7. (a) MRM of WN1 SSW-based composites of 500-hPa geopotential height anomalies (contour interval 10 gpm) in the [-10, 0]-day period before events for the comparison (1979-2012) period. Only statistically significant anomalies at the 95% confidence level of the same sign (Monte-Carlo test) in at least 66.7% of all reanalyses are plotted. (b) Standard deviation of the reanalyses with respect to the MRM divided by the square root of the number of reanalyses for WN1 SSWs (contour interval is 1 gpm). (c) Same as (a) but for the WN1 SSWs minus WN2 SSWs differences of MRM composites of 500-hPa geopotential height anomalies. (d) and (e) Same as (a) and (b) but for WN2 SSWs, respectively. (f) Same as (c) but for displacement-minus-split events.**





**Figure 8. (a-c)** MRM of blocking frequency for the [-10, 0]-day period before the central date of WN1 SSWs of the comparison period (1979-2012) for the: (a) anomaly, (b) absolute, (c) mixed method. The blocking frequency is expressed as the percentage of time (over the 11-day period) during which a blocking was detected at each grid point. Vertical (horizontal) hatching denotes regions where at least **66.7%** of the reanalyses show a significant increase (decrease) of the frequency with respect to the climatology at the 90% confidence level. **(d-f)** Same as (a-c) but for WN2 SSWs. **(g-i)** MRM of the mean blocking frequency in 1000 Monte Carlo trials of 11-day intervals preceding all SSWs dates of the comparison period. In each trial, a set of 11-day intervals prior to the SSWs dates of random years is averaged, so that we obtain a pseudo-climatology of the blocking frequency in the same winter moments as when the SSWs took place. This method avoids any effects of the seasonal cycle of the blocking activity during the extended winter (NDJFM) that would affect the result if we averaged directly the blocking activity during that season.



**Figure 9. Same as Fig. 7 but for MSLP and the [5,35]-day period after SSWs. Contour interval is 0.5 hPa for MRM composites and differences and 0.1 hPa for the standard deviation of the reanalyses.**