# Peer review of "On the representation of major stratospheric warmings in reanalyses"

_Atmospheric Chemistry and Physics, 2019_

## Referee Comment (RC1) · Anonymous Referee #1 · 16 Apr 2019

Review of 'On the representation of major stratospheric warmings in reanalyses' by Ayarzaguena
et al. (2019)

This paper has examined how different features of SSWs (e.g., magnitude, precursors, surface impact)
vary across different reanalysis datasets in both the historical period (1958-1978) ad post-satellite
era. The authors have also examined the differences in features between wavenumber 1 and wavenumber 2
SSWs. The paper is overall interesting and is a valuable contribution to the literature; using historical
data back before 1979 will be useful for the SSW community and this study suggests that despite the
discrepancies between the pre and post satellite era data, the characteristics of SSWs in different
reanalyses act fairly similarly. All of my comments below are minor and hence I suggest only minor
revisions.

One thing to note is that this review was not so convenient to write because of the line numbering. I
have included a line number and a page number for each comment as it appears that the numbers ran to
35 before restarting over again continuously!

Specific Comments:

Line 20; Can you just confirm whether by the 'surface fingerprint', you mean either the
downward impact following the SSW, or the near-surface precursors?
Lines 30-32; It is worth mentioning here that SSWs are not always preceded by precursory
wave activity in the troposphere (most recently for instance, Birner and Albers 2017,
SOLA; White et al. 2019, J.Clim both found that ~30% of SSWs are preceded by lower-
tropospheric wave activity in observations and in a GCM, respectively). I don't mean for
you to go into details regarding this, but it would be good to mention that sometimes the
source of the anomalous wave activity is in the stratosphere. Another good citation to add
would be Garfinkel et al. (2010), J. Clim who found that a deepened Aleutian Low leads to
enhanced upward wave-1 flux. In this part of the text you have only mentioned about blocking
highs preceding SSWs, when many SSWs are preceded by such an anomalously-deep Aleutian Low.
Line 9, page 2; this line suggests that all SSWs impact the tropospheric circulation when in

reality, not all do, and only in the composite mean is there an aggregate impact. It would be better
to make this clearer.
Lines 9-12, page 4; How are SSWs in each reanalysis determined to be 'common'? What is the time
window around the actual SSW date in one reanalysis for which an occurence of a wind reversal in
another reanalysis is deemed to be the same date? You just mention here that four out of seven
reanalyses in the common period must show the same SSW event; but, how is the same event determined?
Line 25, page 4; Can you be clearer here? It is not immediately clear how you chose the common SSWs
to be either D or S here. Did you check each common SSW in each renalysis and then determine if the
majority of reanalyses showed either a D or an S? Or was there some other way?
Line 27, page 4; how sensitive are the results to different levels and latitudes? A sentence or two
would be good to describe the sensitivity. Also, was the 200m difference threshold arbitrarily chosen?
Line 1, page 7; can you better explain how these histograms are calculated? It seems to me that for
each date on the x-axis you take a 21-day window (centred on that date), and count how many SSWs
occurred in that window. You then moved on to the next date and did the same. Is this correct? If so,
it seems to me that by doing this, SSWs are counted multiple times and the histogram may not be a fair
representation. What happens when this window is shortened from 21 days? Shortening the window length
will no doubt be a more accurate way to do this. Just creating bar charts of the #SSWs in each month
would be a fairer and less-ambiguous representation and then just compare the distributions.
In terms of the histograms, it would be useful to test the significance between the individual
histograms using a Kolgomorov-Smirnov test. My guess is that they are significantly different in (a),
but not in (b).
Line 14, page 8; how does the HF look below 100hPa? Say down to 300hPa? Are there any significant anomalies?
Between 300hPa and 100hPa is the comunication region for stratosphere-troposphere coupling that de la Camara
et al. (2017) suggested to be particularly important. 100hPa is already in the stratosphere at high latitudes,
and hence, 300hPa may be a better measure of the upward propagation of wave activity from the troposphere.
Lines 23-24, page 8; This is an interesting result. Is the correct interpretation
that prior to lag -5, the wave activity grows in the stratosphere via constructive interference with the
climatological planetary waves, whereas from lags -5 to 0, anomalous wave growth

occurs? I am wondering if this
is indicative of the Plumb (1981), JAS idea of self-tuning resonance? i.e., a standing climatological wave and a
transient anomalous wave interact constructively to give a growing-in-amplitude wave in the stratosphere? This
wave then grows to very large amplitude and eventually splits the vortex. This is more of a probing statement,
as I do not know for sure. But some interpretation as to why the earlier lags are dominated by the interference
term and the lags closer to zero are dominated by the anomalous term, would be appreciated here.
Lines 12-13, page 9; How sensitive are the results in this figure to this lag window? I ask because the lag
window you have chosen is based on figure 5 which only extends down to 100hPa. In figure 7 you present 500hPa.
Do the significant HF anomalies below 100hPa extend further back in time to before lag -10? If so, then this
would suggest increasing the length of the lag window.

Technical Comments:

Line 26, page 2; what is the 'second one' here?
Line 32, page 2; Here seems a good place to start a new paragraph when you start talking about
the aims/methods of this paper.
Lines 3-4, page 3 (top of page); I think you also examined the downward impact of S and D events,
right? Unless you are classifying S and D, and WN1 and WN2 events as the same (although I don't
think you are)
Line 9, page 3; typo. I think you mean: 'The former analyses the momentum budget during SSWs...'
or something to this effect!
Line 23, page 3; did you perform the interpolation yourselves? A sentence or two describing the
method used would be useful - was it a simple linear interpolation? Or something more complex?
Line 32, page 3; Just to clarify, the anomalies are calculated as the departure of the field from the
daily climatology for EACH reanalysis? Or do you mean the anomaly from the daily climatology over
ALL reanalysis products (i.e., away from the MRM)?
Line 33, page 4; why is the 1981-2010 baseline used instead of the full 1979-2012 period?
Line 8, page 4; imposing --> requiring. Also, I think the Charlton and Polvani (2007) paper must
be cited here! As this is, as I recall, the definition from their paper exactly.
Line 24, page 4; Perhaps better would be: '...with respect to the occurrence of an SSW, according
to the definition in section 2.2'

Line 4, page 6; 'two-folded' --> 'two fold'
Lines 22-24, page 6; somewhere it should be mentioned that only the historical period is considered
in figure 1.
Line 6, page 7; What is meant here by 'traced back to the PNJ'? You haven't previously explicitly
calculated the PNJ (which from section 2.4 I understand to be the difference in wind strength prior to
and following the SSW central date). Are you here referring to the PNJ as just the strength of the U at
60N and 10hPa as shown in figure 1c? If so, then the PNJ as defined in section 2.4 needs to be better
articulated.
Line 23, page 7; I think you mean to compare Fig 3,b,c with Fig 3,f,g?
Line 13, page 8; is this area-averaged? i.e., weighted by the cos(lat)?
Lines 23-24, page 8; 'precedent' --> 'preceding'.
Line 28, page 8; Change to 'historical period'
Figures 4-6; Negative contours would be easier to identify if they were dashed rather than solid. This is
particularly true if there is no significance (and hence no shading)!
Figure 7, caption; Only gridpoints with stat sig values are shaded right? The contours are the full anomalies?
If so, line 4 on page 25 needs to be updated (i.e., change 'plotted' for 'shaded') as it is not clear.
Further, the density of anomaly contours is very high, especially considering that much of the plots are
insignificant. Seeing as the WN1 and WN2 climatological centres of action are important in your description,
it would be useful to put one or two contours (say, in green) for each centre on the plot. Hence, I suggest
to reduce the density of anomaly contours and to just put a couple of contours representing the climatology,
which should not clog up the plot.
Line 21, page 9; 'MMR' is meant to be MRM?
Line 8, page 10; so the bottom row should equal the sum of the top two rows? Further, are the units of the colorbar
percentages?
Line 9, page 10; 'al' --> 'all'
Line 19, page 10; 'non-significant' --> 'insignificant'
Line 7, page 12; 'but at much less extent' --> 'to less of an extent,'
Line 32, page 12, change to 'pre- and post satellite eras.'

---

## Referee Comment (RC2) · Daniela Domeisen (Referee) · 1 May 2019

REVIEW of "On the representation of major stratospheric warmings in reanalyses" by Ayarzagüena et al.

SUMMARY: This paper discusses the representation of SSW events in different reanalysis products. This is an important contribution given the increased use of SSWs for long-range prediction of surface quantities, which are often initialized from and compared against different reanalysis products. This is a timely contribution for the S-RIP project of comparing reanalysis products for the stratosphere.

OVERALL ASSESSMENT: The paper is well written and addresses an interesting and worth-while problem. I have some comments that I hope will improve the manuscript,

see below.

SPECIFIC COMMENTS:

Page 1:

Line 22: "surface fingerprint": does this refer to the signature after the SSW event? Please specify.

Line 26: "lead to": this is not a causal effect, but effects that are linked through thermal wind balance

Line 31 – 34: The literature is rather split about this issue, see e.g. Birner & Albers 2017, Sjoberg & Birner, 2014.

Page 3:

Line 12: "analyzes the SSWs the momentum budget": unclear

Page 4:

Lines 24 – 28: since K. Shibata is a co-author, it would help to clarify the algorithm used in the manuscript in case it's not (yet) published.

Page 5:

Line 25: anomalies from climatology?

Line 17: The deviation in the results of NCEP from other reanalysis products is not surprising. There's an artificial trend in the stratosphere – we found it in Badin & Domeisen, 2014 (pages 1498/1499). I could imagine there's also an S-RIP publication that documents this problem?

Page 6/7:

I'm wondering if it would be helpful to list the classification for all events, not just the ones that are common

Page 7:

Line 8: "can be traced back to the PNJ": this does not sound like an explanation, rather a symptom

Lines 15/16: given the large uncertainties in the pre-satellite period this is difficult to state. However, there are indeed changes in decadal variability of SSW frequency in Domeisen, 2019, JGR, maybe this is helpful?

Page 8:

Lines 1-6: maybe it would be helpful to indicate the changes in stratospheric representation btw the different NCEP reanalysis tools, or maybe refer to the Hitchcock, 2019 paper?

Lines 24 – 26: yes, indeed, this is why it is so difficult to trace waves from the troposphere to the stratosphere. This is not so counterintuitive given the literature on the stratospheric contribution to SSWs.

Line 29: at which level?

Page 11 / Figure 7 / Page 23, line 31: are these differences significantly different from each other? i.e. not just significantly different from climatology?

MINOR COMMENTS:

Page 1:

Line 30: I would suggest using Charlton et al (2007) as the authoritative reference here.

Page 2:

Line 8: Martius et al (2009) seems like the perfect reference here, it's already included in a different place in the manuscript

Lines 10 – 16: would it make sense to include the classification into reflective and

absorptive events here (Kodera et al, 2016)?

Line 18: given the very limited number of studies of stratospheric effects on the ocean I would not call the assessment of oceanic phenomena based on the stratosphere a "common metric"

Line 21: leave out "interestingly", and "largely"

Line 22: "assimilation data sources": do you mean the data used for the assimilation of observational data into the reanalysis products?

Line 27: "than in the second one". Do you mean "than during the satellite era"?

Page 3:

Line 6: is made on > is given to

Line 26: do you mean "across different reanalysis products"?

Page 4:

Line 29: "similarly": do you mean the identification was similar or it was also included in the table?

Page 5:

Line 28: I'm not sure what is meant by "discrepancies" (also: page 6, line 14)

Page 8:

Line 9: ones -> SSWs

line 19: "reanalysis deviation": not clear what this means

Lines 23 – 26: be more clear which terms this corresponds to in the equation

REFERENCES: Badin, G. and D.I.V. Domeisen (2014): A search for chaotic behavior in Northern Hemisphere stratospheric variability, Journal of the Atmospheric Sciences,

Vol. 71 (4), 1494 - 1507. doi: 10.1175/JAS-D-13-0225.1.

Birner, T., & Albers, J. R. (2017). Sudden Stratospheric Warmings and Anomalous Upward Wave Activity Flux. Sola, 13A(Special_Edition), 8–12. http://doi.org/10.2151/sola.13A-002

Charlton, A., Polvani, L., Perlwitz, J., & Sassi, F. (2007). A new look at stratospheric sudden warmings. Part II: Evaluation of Numerical Model Simulations. Journal of Climate

Domeisen, D. I. V. (2019). Estimating the Frequency of Sudden Stratospheric Warming Events from Surface Observations of the North Atlantic Oscillation. Journal of Geophysical Research-Atmospheres, 1–38. http://doi.org/10.1029/2018JD030077

Kodera, K., Mukougawa, H., Maury, P., Ueda, M., & Claud, C. (2016). Absorbing and reflecting sudden stratospheric warming events and their relationship with tropospheric circulation. Journal of Geophysical Research-Atmospheres, 121(1), 80–94. http://doi.org/10.1002/2015JD023359

Sjoberg, J.P., and Birner, T. (2014): Stratospheric Wave–Mean Flow Feedbacks and Sudden Stratospheric Warmings in a Simple Model Forced by Upward Wave Activity Flux. J. Atmos. Sci.

---

## Author Response (AR1)

Dear editor,

Please find enclosed our point-by-point response to the referees (they are the same as uploaded as Author Comments (AC1 and AC2)). They contain the referees' comments (in blue) and our replies to their questions and, if appropriate, information on the relevant changes introduced in the revised version the manuscript (in black).

After the point-to-point response, you will find the revised version of the manuscript and the supplementary material with all changes highlighted in blue (added text) and red (deleted text) colors.

Best regards,

Blanca Ayarzagüena and co-authors.

**Review of 'On the representation of major stratospheric warmings in reanalyses' by Ayarzagüena et al. (2019)**

This paper has examined how different features of SSWs (e.g., magnitude, precursors, surface impact) vary across different reanalysis datasets in both the historical period (1958-1978) ad post-satellite era. The authors have also examined the differences in features between wavenumber 1 and wavenumber 2 SSWs. The paper is overall interesting and is a valuable contribution to the literature; using historical data back before 1979 will be useful for the SSW community and this study suggests that despite the discrepancies between the pre and post satellite era data, the characteristics of SSWs in different reanalyses act fairly similarly. All of my comments below are minor and hence I suggest only minor revisions.

Thanks for your comments. They have been very useful and have improved our manuscript.

One thing to note is that this review was not so convenient to write because of the line numbering. I have included a line number and a page number for each comment as it appears that the numbers ran to 35 before restarting over again continuously!

We apologize for this inconvenience. Before the submission we did not realize that the word template had set up by default the line number restarting at each page. We have fixed this problem.

Specific Comments:
Line 20; Can you just confirm whether by the 'surface fingerprint', you mean either the downward impact following the SSW, or the near-surface precursors?
We mean the downward impact after the SSW. We have replaced "fingerprint" for "response" to clarify it.

Lines 30-32; It is worth mentioning here that SSWs are not always preceded by precursory wave activity in the troposphere (most recently for instance, Birner and Albers 2017, SOLA; White et al. 2019, J.Clim both found that ~30% of SSWs are preceded by lower tropospheric wave activity in observations and in a GCM, respectively). I don't mean for you to go into details regarding this, but it would be good to mention that sometimes the source of the anomalous wave activity is in the stratosphere.
Thanks for the suggestion. First, we would like to indicate that we were not referring to an enhancement of lower tropospheric wave activity in this part, but just tropospheric wave activity at any level, in the upper troposphere too. Nevertheless, we acknowledge that recent studies have shown that the enhancement of wave activity often happens within the stratosphere and/or is related to a preconditioning of the mean stratospheric flow. We have included a comment about this in the introduction section (new Lines 32-34)

Another good citation to add would be Garfinkel et al. (2010), J. Clim who found that a deepened Aleutian Low leads to enhanced upward wave-1 flux. In this part of the text you have only mentioned about blocking highs preceding SSWs, when many SSWs are preceded by such an anomalously-deep Aleutian Low.
Thanks for the suggestion. In the original version (Section 4.2), we already referred to the anomalously deep Aleutian low as a precursor of SSWs and included Garfinkel et al (2010)'s citation. In the revised text, we have also mentioned it in Line 36 of the Introduction.

Line 9, page 2; this line suggests that all SSWs impact the tropospheric circulation when in reality, not all do, and only in the composite mean is there an aggregate impact. It would be better to make this clearer.
We have included some clarifications about the uncertainty about the tropospheric response to SSWs in new lines 45-48.

Lines 9-12, page 4; How are SSWs in each reanalysis determined to be 'common'? What is the time window around the actual SSW date in one reanalysis for which an occurrence of a wind reversal in another reanalysis is deemed to be the same date? You just mention here that four out of seven reanalyses in the common period must show the same SSW event; but, how is the same event determined?
The list of common SSWs has been provided by Amy Butler via the S-RIP initiative (https://www.sparc-climate.org/activities/reanalysis-intercomparison/). For that classification, the events were first individually identified in each reanalysis based on the reversal of the zonal mean zonal wind at 60ºN and 10hPa from November to March, and additional restrictions to ensure the independence between events and the exclusion of stratospheric final warmings (Charlton and Polvani, 2007). Secondly, the number of reanalyses that identify

an event around the same date was determined. It was not necessary to impose any condition to determine whether events detected by different reanalyses were or not the same, because the spread across reanalyses in the dates of SSWs is very small (typically within one or two days). Only an event in the historical period (17 December 1965) showed a difference of more than a week between NCEP/NCAR and the other two historical reanalyses (JRA-55 and ERA-40). In that case, the date of the common event was computed as the average of the dates for the latter reanalyses (those with more vertical levels in the stratosphere) (Chapter 6 of S-RIP initiative). Finally, common SSWs are those identified by at least two of the three reanalyses in the historical period and by at least four out of seven reanalyses in the comparison period. We have clarified this in the new version of the manuscript (lines 116-118).

Line 25, page 4; Can you be clearer here? It is not immediately clear how you chose the common SSWs to be either D or S here. Did you check each common SSW in each renalysis and then determine if the majority of reanalyses showed either a D or an S? Or was there some other way?
Yes, it was exactly done as the reviewer indicated. Again this follows the S-RIP initiative guidelines. We have clarified it in the text (lines 134-137).

Line 27, page 4; how sensitive are the results to different levels and latitudes? A sentence or two would be good to describe the sensitivity. Also, was the 200m difference threshold arbitrarily chosen?
The methodology applied here corresponds to that described by Barriopedro and Calvo (2014), which is based on the algorithm previously presented by Bancalà et al (2012). The latter used data at 10 and 50 hPa, while Barriopedro and Calvo (2014) used the 50 hPa level only because: i) this level is close to that of the maximum amplitude of climatological WN2 and not far from that of WN1; ii) some reanalyses (e.g. NCEP/NCAR) have their model tops at 10 hPa, which may introduce artificial biases. Still, Barriopedro and Calvo (2014) already compared their classification with that by Bancalà et al (2012) and obtained very similar conclusions, suggesting that the method is not too sensitive to the chosen levels. This is also supported by: i) the time evolution of composites of the WN1 and WN2 components of anomalous heat flux for WN1 and WN2 SSWs, which show similar signatures at 10 and 50 hPa (Fig. R1.1); ii) the robustness of the SSW classification across reanalyses (in contrast to most algorithms that classify D and S SSW events).

[Figure]

**Figure R1. 1. (a)** Time evolution of MRM of WN1 (blue) and WN2 (red) component of anomalous heat flux (K m s$^{-1}$) at 50hPa (solid line) and 10hPa (dash line) from -30 days to 30 days after the occurrence of WN1 SSWs. **(b)** Same as (a) but for WN2 SSWs.

Regarding the sensitivity to different latitudes, both studies used 60ºN for the classification of WN1 and WN2 SSWs. This latitude band is close to the maximum amplitude of both climatological WN1 and WN2 waves (Fig. R1.2), and it is also where the reversal of the zonal mean zonal wind is computed for the identification of SSWs.
As for the threshold of $Z_2-Z_1$, it was not chosen arbitrarily. The 200 m corresponds to the 90$^{th}$ percentile of the difference of WN2 minus WN1 components of the geopotential height at 50 hPa, as indicated by Barriopedro and Calvo (2014).

Given that this is a published algorithm, and it has also been included in the ongoing S-RIP report, we have decided not to provide more information in our manuscript. Nevertheless, we have now explicitly referred to Barriopedro and Calvo (2014) for more details (lines 142-143).

[Figure]

**Figure R1. 2:** (a) Multi-reanalysis mean of the climatological WN1 component of geopotential height at 50hPa in January and February (Contour interval: 30m). (b) Same as (a) but for the climatological WN2 component (Contour interval: 20m).

Line 1, page 7; can you better explain how these histograms are calculated? It seems to me that for each date on the x-axis you take a 21-day window (centred on that date), and count how many SSWs occurred in that window. You then moved on to the next date and did the same. Is this correct? If so, it seems to me that by doing this, SSWs are counted multiple times and the histogram may not be a fair representation. What happens when this window is shortened from 21 days? Shortening the window length will no doubt be a more accurate way to do this. Just creating bar charts of the #SSWs in each month would be a fairer and less-ambiguous representation and then just compare the distributions.

Yes, the procedure described by the reviewer to create the seasonal distribution of SSWs is correct. Additionally, a 10-day running mean was applied to smooth the distribution. Similar approaches have also been used previously (e.g. Gómez-Escolar et al. 2012). We have explained more carefully the way we computed this seasonal distribution of SSWs (lines 203-205).

We do not totally agree with the reviewer on the use of a monthly histogram of SSWs. The histograms are very useful for giving a brief overview of the monthly frequency of SSWs. However, in our specific case we think that it makes more sense the use of consecutive bins that overlap to build the distribution. The total mean frequency of SSWs has already been shown in Table 2, and in this part of the study we are not interested anymore in the exact number of SSWs but in their distribution along winter. In particular, we would like to know if reanalyses present important differences in their distributions, i.e. if the SSWs captured by each reanalysis correspond to the same events in the other datasets. In this sense, the division per calendar months is somehow arbitrary and might lead to artificial differences between reanalyses. For instance, if a SSW occurred by the turn of a month, it might be detected on the very first days of a month in some reanalyses and on the very last days of the previous month in other datasets. As such, the typical monthly histogram would prevent from knowing if they are the same event or not. This problem is avoided with our approach. Gómez-Escolar et al. (2012) already showed that the bimodal distribution of SSWs could be missed in monthly histograms.

Regarding the sensitivity of results to window width, we have shortened this window as suggested by the reviewer. For instance, Figure R1.3 presents the same distribution but for 11-day windows (± 5 days). We also include in Figure R1.4 the results for a 21-day window to enable the comparison of results herein. We can see that the main conclusions do not change. We still detect the shift of SSWs to a later date in the comparison period, the good agreement between reanalyses in that period and the closer resemblance of distributions between ERA-40 and JRA-55 than between any of these two and the NCEP-NCAR reanalysis in the historical era. Despite the agreement in results, we prefer to keep the 21-day window width (± 10 days), because it is closer to a month and so, it makes easier to identify the main peaks of SSWs in each period than a shorter window.

Please also note that Figure 2 has been slightly modified in the revised manuscript as we now represent the number of SSWs per decade instead of the total number of events. Although the shape of the distribution does not change, it allows a more straight-forward comparison of results between the historical and comparison periods, which have different lengths.

[Figure]

**Figure R1. 3:** SSW total frequency distribution within ±5 day periods from the date displayed in the x-axis for: (a) the historical period (1958-1978) and (b) the comparison period (1979-2012). Time series are smoothed with a 10-day running mean.

[Figure]

**Figure R1. 4.** Same as Figure R1.3 but SSW total frequency distribution within ±10 day periods

In terms of the histograms, it would be useful to test the significance between the individual histograms using a Kolgomorov-Smirnov test. My guess is that they are significantly different in (a), but not in (b).
We have applied a two-sample Kolmogorov-Smirnov test. As the reviewer expected, the distributions of SSWs in the comparison period are indistinguishable between each other and statistically significantly different from those in the historical one at the 99% confidence level. In contrast, in the historical period the NCEP-NCAR distribution is significantly different from those of JRA-55 and ERA-40 (p < 0.01) according to the same test.

The SSW distribution of JRA-55 and ERA-40 are still indistinguishable in this period. The same results are found when shortening the time window of the distribution to 11 days.
We have included this information in the manuscript (lines 206-209 and 219-220)

Line 14, page 8; how does the HF look below 100hPa? Say down to 300hPa? Are there any significant anomalies? Between 300hPa and 100hPa is the communication region for stratosphere-troposphere coupling that de la Camara et al. (2017) suggested to be particularly important. 100hPa is already in the stratosphere at high latitudes, and hence, 300hPa may be a better measure of the upward propagation of wave activity from the troposphere.
According to the reviewer's suggestion, we have repeated the analysis up to 300hPa. However, our conclusions remain the same given that the region with the strongest signal is above 100hPa. Nevertheless, significant values are also observed between 300 and 100 hPa in most cases supporting a stratosphere-troposphere coupling in the multi-reanalysis mean. In the revised manuscript, we have updated figures 4, 5 and 6 by extending them down to 300hPa.

Lines 23-24, page 8; This is an interesting result. Is the correct interpretation that prior to lag -5, the wave activity grows in the stratosphere via constructive interference with the climatological planetary waves, whereas from lags -5 to 0, anomalous wave growth occurs? I am wondering if this is indicative of the Plumb (1981), JAS idea of self-tuning resonance? i.e., a standing climatological wave and a transient anomalous wave interact constructively to give a growing-in-amplitude wave in the stratosphere? This wave then grows to very large amplitude and eventually splits the vortex. This is more of a probing statement, as I do not know for sure. But some interpretation as to why the earlier lags are dominated by the interference term and the lags closer to zero are dominated by the anomalous term, would be appreciated here.
We thank the reviewer for this comment. We prefer though not to include this reflection in the mentioned lines. The results that the reviewer is referring to correspond to Figure 5 where all SSWs of the comparison period are considered. However, when separating WN1 and WN2 SSWs (Fig. 6), we can see that their respective peaks of anomalous HF come from different dynamical forcings and occur in different timing. Whereas WN1 SSWs are mainly dominated by persistent but moderate anomalous HF originated from the constructive interference between anomalous and climatological planetary waves during 20 days, WN2 SSWs are preceded by a strong and short pulse of HF due to anomalous waves only in the last five days prior to the SSW onset. Thus, it does not seem that the mechanism suggested by the reviewer is clearly working for none of SSW types. Moreover, it seems that the suggested interpretation should be more likely true for the WN2 events than for the WN1 ones, as most of WN2 SSWs have associated a vortex split. However, in that case, we can only identify a strong anomalous burst of wave activity in the 5 days prior to the SSW occurrence.
Nevertheless, the reviewer's comment was very useful for us and has been used in the following Section 4.2 when discussing the tropospheric circulation anomalies preceding WN2 SSWs. The spatial coincidence of these anomalies and the anti-nodes of the climatological WN2 wave would suggest that the constructive interference in the troposphere is important prior to WN2 SSWs, even if the previous results on heat flux anomalies at higher levels rule out the relevance of the wave interference for these events. In the revised version (lines 312-317), we have tried to solve this apparent contradiction by including the idea of a self-tuning resonance of waves in the stratosphere as a result of a slight enhancement of tropospheric wave activity, probably due to the linear interference of waves. As shown by Albers and Birner (2014), this resonance would be more likely when the polar vortex is preconditioned in an initial structure close to its resonant point as it happens in the case of WN2 events.

Lines 12-13, page 9; How sensitive are the results in this figure to this lag window? I ask because the lag window you have chosen is based on figure 5 which only extends down to 100hPa. In figure 7 you present 500hPa. Do the significant HF anomalies below 100hPa extend further back in time to before lag -10? If so, then this would suggest increasing the length of the lag window.
As the reviewer indicates, the selection of the window (-10, 0) day was based on Figure 5 and in particular, the peak of anomalous eddy heat flux above 100hPa. When extending the new Figure 5 down to 300hPa, we see that the significant HF anomalies below 100hPa do not extend beyond lag -10, supporting our choice. In addition, we have repeated the analysis for two wider time windows: (-20,0) and (-15,0) days (Fig. R1.5 and R1.6, respectively), and the results do not change substantially.
Another point to highlight is that this 10-day window has been very commonly used in previous analyses of the upward branch of the troposphere-stratosphere coupling (e.g.: Martius et al., 2009; Nishii et al., 2011;

Ayarzagüena et al., 2015), as it corresponds to the approximate time that planetary waves take to propagate from the troposphere to the stratosphere (Limpasuvan et al. 2004).

Given that our main results are not sensitive to the width of the time window considered in Figure 7 and based on the previous literature we prefer to keep the (-10, 0) interval. Nevertheless, we have added a short comment justifying more in detail the selection of lag windows in the revised text (lines 297-299).

**Z500 [-20, 0]-day period**

**Figure R1. 5.** (a) MRM of WN1 SSW-based composites of 500-hPa geopotential height anomalies (contour interval 20 gpm) in the [-20, 0]-day period before events for the comparison (1979-2012) period. Only statistically significant anomalies at the 95% confidence level of the same sign (Monte-Carlo test) in at least 66.7% of all reanalyses are shaded. (b) Standard deviation of the reanalyses with respect to the MRM divided by the square root of the number of reanalyses for WN1 SSWs (contour interval is 1 gpm). (c) Same as (a) but for the WN1 SSWs minus WN2 SSWs differences of MRM composites of 500-hPa geopotential height anomalies. Shading denotes statistically significant differences at the 95% confidence level in at least 66.7% of all reanalyses (Monte-Carlo test). (d) and (e) Same as (a) and (b) but for WN2 SSWs, respectively. (f) Same as (c) but for displacement-minus-split events. Green contours in (a) and (d) show the MRM climatological WN1 and WN2 of 500-hPa geopotential height from November to March, respectively (contours: ±40 and ±80 gpm).

[Figure]

**Figure R1. 6:** Same as Figure R1.5 but for the period [-15,0]-day period before SSWs.

Technical Comments:
Line 26, page 2; what is the 'second one' here?
The post-satellite period. We have modified it.

Line 32, page 2; Here seems a good place to start a new paragraph when you start talking about the aims/methods of this paper.
It was actually a new paragraph, although it did not look like that. After the inclusion of a new word, the separation between the two paragraphs is clearer.

Lines 3-4, page 3 (top of page); I think you also examined the downward impact of S and D events, right? Unless you are classifying S and D, and WN1 and WN2 events as the same (although I don't think you are)
Yes, we have also examined the downward impact of S and D events. In the revised text, we have listed both classifications (which are independent).

Line 9, page 3; typo. I think you mean: 'The former analyses the momentum budget during SSWs…' or something to this effect!
Yes, we have corrected it.

Line 23, page 3; did you perform the interpolation yourselves? A sentence or two describing the method used would be useful - was it a simple linear interpolation? Or something more complex?
The models of the reanalyses included in the study have different horizontal resolutions and provide output on different grids. If the output on the 2.5°x2.5° grid was available, we just used it. When this was not possible (only NCEP-CFSR and NASA-MERRA), we used the cdo tool remapcon that performs a first order conservative remapping of the input fields.
Both reanalyses perform very well when comparing with the rest of datasets, and given that we applied the same algorithm in the calculation, we do not think remapping has any effect on the SSW-related computations. We have briefly included all this information in the revised text (new lines 96-97).

Line 32, page 3; Just to clarify, the anomalies are calculated as the departure of the field from the daily climatology for EACH reanalysis? Or do you mean the anomaly from the daily climatology over ALL reanalysis products (i.e., away from the MRM)?

The anomalies are computed as the departure of the field from the daily climatology of each reanalysis. As we are assessing the performance of reanalyses related to anomalous fields, we think it makes more sense to compute the anomalies in each reanalysis as departures from its own climatology. In this case, any bias in the mean flow that does not contribute to the anomalous behavior is removed.
We have now specified "of each reanalysis" to make it clear.

Line 33, page 4; why is the 1981-2010 baseline used instead of the full 1979-2012 period?

This was one of the recommendations of the S-RIP initiative. It also corresponds to the period that NOAA is currently considering for computing the climatology, based on the WMO indications about the computation of climatological values from 30-yr averages (https://www.ncdc.noaa.gov/data-access/land-based-station-data/land-based-datasets/climate-normals/1981-2010-normals-data). Moreover, this 30-yr baseline matches with the full 1979-2012 period, excluding only two years before and after.
Due to the shortness of the historical period, it was not possible to consider any 30-yr period and so, we used the full period as a baseline.
We have not included any clarification in the text since the 1981-2010 baseline is the typical period currently used in many studies.

Line 8, page 4; imposing --> requiring. Also, I think the Charlton and Polvani (2007) paper must be cited here! As this is, as I recall, the definition from their paper exactly.

Modified

Line 24, page 4; Perhaps better would be: '...with respect to the occurrence of an SSW, according to the definition in section 2.2'

Included

Line 4, page 6; 'two-folded' --> 'two fold'

Corrected

Lines 22-24, page 6; somewhere it should be mentioned that only the historical period is considered in figure 1.

We have mentioned it now when referring to the differences between reanalyses in the standard deviation of polar temperature and zonal wind at 10hPa (lines 211).

Line 6, page 7; What is meant here by 'traced back to the PNJ'? You haven't previously explicitly calculated the PNJ (which from section 2.4 I understand to be the difference in wind strength prior to and following the SSW central date). Are you here referring to the PNJ as just the strength of the U at 60N and 10hPa as shown in figure 1c? If so, then the PNJ as defined in section 2.4 needs to be better articulated.

Based on the reviewers' comments, we have modified the expression "traced back to the PNJ". In the revised version, we have only related the different SSW distribution to a different climatological PNJ for NCEP/NCAR and the other two reanalyses in the historical period.
Regarding the issue about the PNJ definition, there must have been a misunderstanding. In Section 2.4 we defined the deceleration of the PNJ during SSWs (decelu) as the difference in the strength of the zonal wind at 60ºN and 10hPa before and after the SSW, but not the PNJ itself, which should therefore be understood as the strength of the zonal wind at 60ºN and 10 hPa. .
To avoid confusion, a reference to the Figure 1c has been included in the text when mentioning the climatology of the PNJ (lines 214). We have also removed the acronym (PNJ) in Section 2.4, where we also mentioned decelu, as it could cause some misunderstanding.

Line 23, page 7; I think you mean to compare Fig 3,b,c with Fig 3,f,g?

Yes, thanks

Line 13, page 8; is this area-averaged? i.e., weighted by the cos(lat)?

Yes. We have included area-averaged.

Lines 23-24, page 8; 'precedent' --> 'preceding'.
Corrected

Line 28, page 8; Change to 'historical period'
Changed

Figures 4-6; Negative contours would be easier to identify if they were dashed rather than solid. This is particularly true if there is no significance (and hence no shading)!
Done

Figure 7, caption; Only gridpoints with stat sig values are shaded right? The contours are the full anomalies? If so, line 4 on page 25 needs to be updated (i.e., change 'plotted' for 'shaded') as it is not clear.
Yes, the reviewer is correct. We have corrected the caption.

Further, the density of anomaly contours is very high, especially considering that much of the plots are insignificant. Seeing as the WN1 and WN2 climatological centres of action are important in your description, it would be useful to put one or two contours (say, in green) for each centre on the plot. Hence, I suggest to reduce the density of anomaly contours and to just put a couple of contours representing the climatology, which should not clog up the plot.
We thank the reviewer for this comment. It has indeed improved the quality of Figure 7. We have realized that the density of anomaly contours in Figure 9 was also too high and have reduced it.

Line 21, page 9; 'MMR' is meant to be MRM?
Yes. Changed

Line 8, page 10; so the bottom row should equal the sum of the top two rows?
Not exactly. The bottom row of Figure 8 corresponds to a "climatology" of the mean blocking frequency prior to SSWs. This was computed as the mean blocking frequency in 1000 Monte Carlo trials of 11-day intervals preceding all SSW dates of the comparison period (note that the date of occurrence varies from case to case). In each trial, a set of 11-day intervals prior to the SSWs dates but with random years is averaged, so that we obtain a pseudo-climatology of the blocking frequency in the same winter periods as when the SSWs took place. This method avoids any effect of the seasonal cycle in blocking activity during the extended winter (NDJFM) that would affect the result. It also provides a fair comparison with the two top rows, which contain the same number and calendar chunks of the winter season as the bottom figure but with the actual SSW dates. The sum of the top two rows is expected to be different to the bottom figure, as there are not SSWs every year. The above description was already included in the caption of Figure 8 in the original version. We still prefer to leave this description in the caption. Inserting it in the main text would make the discussion of the results more tedious. Nevertheless, we have slightly modified the text to avoid the confusion highlighted by the reviewer (lines 331-333). We have also added a reference in the main text to the details given in the figure caption.

Further, are the units of the colorbar percentages?
Yes. We have specified the units in the text (line 328) and the caption of Figure 8.

Line 9, page 10; 'al' --> 'all'
Sorry, we could not find this typo.

Line 19, page 10; 'non-significant' --> 'insignificant'
We do not fully agree with the reviewer in this point. We are just applying a statistical test to determine the significance of results at a given confidence level, but this test does not mean that the result is not statistically significant at another confidence level. Moreover, although the result is not statistically significant, this does not imply that it is negligible or not meaningful in a physical sense. Thus, we prefer "non-significant" rather than "insignificant".

Line 7, page 12; 'but at much less extent' --> 'to less of an extent,'
We thank the reviewer for highlighting the mistake. We have corrected the expression to "to a much less extent".

Line 32, page 12, change to 'pre- and post satellite eras.'
We think it is correct as it is.

SUMMARY: This paper discusses the representation of SSW events in different reanalysis products. This is an important contribution given the increased use of SSWs for long-range prediction of surface quantities, which are often initialized from and compared against different reanalysis products. This is a timely contribution for the S-RIP project of comparing reanalysis products for the stratosphere.

OVERALL ASSESSMENT: The paper is well written and addresses an interesting and worth-while problem. I have some comments that I hope will improve the manuscript, see below.

We thank the reviewer for the useful comments that have contributed to improve the manuscript. Please see below our replies in blue color

SPECIFIC COMMENTS:
Page 1:
Line 22: "surface fingerprint": does this refer to the signature after the SSW event? Please specify.
Yes, it does. We have replaced "fingerprint" for "response" to clarify it.

Line 26: "lead to": this is not a causal effect, but effects that are linked through thermal wind balance
We understand what the reviewer means and it is true that both the vertical shear of zonal wind and the meridional temperature gradient are connected through thermal wind balance, so it is not easy to determine what is causing what. However, in the specific case of SSWs, changes in the meridional heat flux are the forcing that leads to changes in the wind. Moreover, in many SSWs it is the change in the polar temperature that precedes the maximum deceleration of the wind. Thus, we think the "lead to" is justified in this case.

Line 31 – 34: The literature is rather split about this issue, see e.g. Birner & Albers 2017, Sjoberg & Birner, 2014.
We agree with the reviewer that recent studies have already shown that the enhancement of wave activity prior to SSWs tends to happen within the stratosphere and/or is related to a preconditioning of the mean stratospheric flow. We have included a comment about this in the Introduction of the revised manuscript (new lines 32-34).

Page 3:
Line 12: "analyzes the SSWs the momentum budget": unclear
We have slightly modified the sentences to make it clear. The new sentence reads like this: "The former analyzes the momentum budget during SSWs restricted to the post-satellite period"

Page 4:
Lines 24 – 28: since K. Shibata is a co-author, it would help to clarify the algorithm used in the manuscript in case it's not (yet) published.
According to the reviewer's suggestion, the description of the algorithm has been extended in the new version (see new lines 128-134).

Page 5:
Line 25: anomalies from climatology?
Yes. We added "daily" to make it clearer.

Page 6:
Line 17: The deviation in the results of NCEP from other reanalysis products is not surprising. There's an artificial trend in the stratosphere – we found it in Badin & Domeisen, 2014 (pages 1498/1499). I could imagine there's also an S-RIP publication that documents this problem?
Thanks for the reference. Unfortunately, most of the S-RIP publications (or even earlier papers) that document the worse performance of NCEP/NCAR in comparison to the other reanalyses are focused on the post-satellite era (e.g.: Manney et al., 2003, Long et al., 2017). In contrast, in this part of the manuscript we are addressing the inter-reanalysis differences in the historical period. We are not aware of other S-RIP publications reporting this issue, and hence we have mentioned the artificial trend in the stratosphere found by Badin & Domeisen

(2014) in the first 50 years of the data record and related that finding to our results by the end of Section 3.1 (lines 214-217) and in the Conclusions (lines 389-390).

Page 6/7: I'm wondering if it would be helpful to list the classification for all events, not just the ones that are common
Thanks for the suggestion. However, we think it is not necessary for different reasons. First, as indicated in the text, most of the differences are more likely due to specific thresholds or methodological issues rather than relevant biases in the reanalyses. In addition, we are using this information in Table 2, as a brief overview of the reanalyses' performance when different events are considered based on fixed criteria. The remaining analyses in the manuscript are based on the events shown in Table 1. As the classification requested by the reviewer is a result of the SRIP initiative (to be included in Chapter 6 of the SRIP report), we have just added an additional reference to that chapter in Section 2 when talking about the classification of SSWs.

Page 7:
Line 8: "can be traced back to the PNJ": this does not sound like an explanation, rather a symptom
We have modified the sentence to avoid confusion. In particular, we have only related the NCEP/NCAR peak of SSWs in early winter to a weaker climatological PNJ in this reanalysis than in the other two.

Lines 15/16: given the large uncertainties in the pre-satellite period this is difficult to state. However, there are indeed changes in decadal variability of SSW frequency in Domeisen, 2019, JGR, maybe this is helpful?
Yes, it certainly helps. We have included the reviewer's comment and some references to previous studies that reported a multi-decadal variability of SSW frequency (including Domeisen 2019). Multi-decadal changes in SSW frequency could also translate to the intra-seasonal distribution of SSWs. Indeed, in the new version of the manuscript, we have confirmed that the SSW distributions of the historical and satellite periods are statistically significant, according to a Kolmogorov-Smirnov test.

Page 8:
Lines 1-6: maybe it would be helpful to indicate the changes in stratospheric representation between the different NCEP reanalysis tools, or maybe refer to the Hitchcock, 2019 paper?
NCEP/NCAR and NCEP-DOE reanalyses are using basically the same model although with different versions, 1995 and 1998, respectively. Most of the improvements made in NCEP-DOE from NCEP/NCAR are related to changes in the lower levels (troposphere), except for the prescription of a new climatology of ozone (Kanamitsu et al., 2002; Long et al., 2017). Other differences in the concentrations of $CO_2$ or radiation schemes might also explain the small differences in results between both NCEP reanalyses.
In the revised manuscript, we have briefly extended the description of differences in the setup and models of both NCEP/NCAR and NCEP-DOE based on Kanamitsu et al. (2002), Fujiwara et al. (2017) and Long et al. (2017) (new lines 245-254).

Page 9:
Lines 24 – 26: yes, indeed, this is why it is so difficult to trace waves from the troposphere to the stratosphere. This is not so counterintuitive given the literature on the stratospheric contribution to SSWs.
Following the recommendations of Reviewer#1, in the revised text we have extended the discussion and inserted references to the recent literature on this topic (lines 312-318). In particular, we have stressed the special importance of the initial state of the polar vortex for the occurrence of WN2 SSWs (e.g. Albers and Birner 2014), the type of events discussed in this part of the study. In those cases, an initial vortex structure close to its resonant point is prone to lead to the split of SSWs with a small increase of tropospheric wave forcing.

Line 29: at which level?
We first checked at 10 and 20hPa, where we found the largest values of anomalous heat flux. However, they are probably not the best levels if we are trying to connect those changes with tropospheric structures. We have removed this sentence from the discussion.

Page 11 / Figure 7 / Page 23, line 31: are these differences significantly different from each other? i.e. not just significantly different from climatology?
Yes, they are. Panel c shows WN1-minus-WN2 differences and the shading indicates that these differences are statistically significantly different from each other. We have corrected the figure caption.

MINOR COMMENTS:
Page 1:
Line 30: I would suggest using Charlton et al (2007) as the authoritative reference here.
Modified

Page 2:
Line 8: Martius et al (2009) seems like the perfect reference here, it's already included in a different place in the manuscript
Included

Lines 10 – 16: would it make sense to include the classification into reflective and absorptive events here (Kodera et al, 2016)?
We prefer to keep it as it is, because we are not referring to these events later on.

Line 18: given the very limited number of studies of stratospheric effects on the ocean I would not call the assessment of oceanic phenomena based on the stratosphere a "common metric"
We just meant just the other way, i.e. oceanic effects on the stratospheric variability. Actually, we were mainly referring to the ENSO effects on the polar stratosphere or other phenomena that have also been recently explored such as PDO or MJO. We have slightly modified the sentence to clarify it.

Line 21: leave out "interestingly", and "largely"
Done

Line 22: "assimilation data sources": do you mean the data used for the assimilation of observational data into the reanalysis products?
Yes. This has been modified

Line 27: "than in the second one". Do you mean "than during the satellite era"?
Yes. We have changed it

Page 3:
Line 6: is made on > is given to
Changed.

Line 26: do you mean "across different reanalysis products"?
We meant across different reanalyses, not products. It has been corrected and clarified.

Page 4:
Line 29: "similarly": do you mean the identification was similar or it was also included in the table?
We meant that the identification was carried out in a similar way as for the common dates. We have clarified it.

Page 5:
Line 28: I'm not sure what is meant by "discrepancies" (also: page 6, line 14)
In the first case we have clarified that it means to the lack of consensus on the precursor role of blockings in SSWs. As for page 6 (now line 184), we have just replaced discrepancies for reanalyses results.

Page 8:
Line 9: ones -> SSWs
Corrected

line 19: "reanalysis deviation": not clear what this means
Differences across reanalyses.

Lines 23 – 26: be more clear which terms this corresponds to in the equation
Done

[revised manuscript text omitted]